# Improving the Expected Improvement Algorithm

**Chao Qin**
Columbia Business School
New York, NY 10027
cqin22@gsb.columbia.edu

**Diego Klabjan**
Northwestern University
Evanston, IL 60208
d-klabjan@northwestern.edu

**Daniel Russo**
Columbia Business School
New York, NY 10027
djr2174@gsb.columbia.edu

## Abstract

The expected improvement (EI) algorithm is a popular strategy for information collection in optimization under uncertainty. The algorithm is widely known to be too greedy, but nevertheless enjoys wide use due to its simplicity and ability to handle uncertainty and noise in a coherent decision theoretic framework. To provide rigorous insight into EI, we study its properties in a simple setting of Bayesian optimization where the domain consists of a finite grid of points. This is the so-called best-arm identification problem, where the goal is to allocate measurement effort wisely to confidently identify the best arm using a small number of measurements. In this framework, one can show formally that EI is far from optimal. To overcome this shortcoming, we introduce a simple modification of the expected improvement algorithm. Surprisingly, this simple change results in an algorithm that is asymptotically optimal for Gaussian best-arm identification problems, and provably outperforms standard EI by an order of magnitude.

## 1 Introduction

Recently Bayesian optimization has received much attention in the machine learning community [21]. This literature studies the problem of maximizing an unknown black-box objective function by collecting noisy measurements of the function at carefully chosen sample points. At first a prior belief over the objective function is prescribed, and then the statistical model is refined sequentially as data are observed. *Expected improvement (EI)* [13] is one of the most widely-used Bayesian optimization algorithms. It is a greedy improvement-based heuristic that samples the point offering greatest expected improvement over the current best sampled point. EI is simple and readily implementable, and it offers reasonable performance in practice.

Although EI is reasonably effective, it is too greedy, focusing nearly all sampling effort near the estimated optimum and gathering too little information about other regions in the domain. This phenomenon is most transparent in the simplest setting of Bayesian optimization where the function's domain is a finite grid of points. This is the problem of best-arm identification (BAI) [1] in a multi-armed bandit. The player sequentially selects arms to measure and observes noisy reward samples with the hope that a small number of measurements enable a confident identification of the best arm. Recently Ryzhov [20] studied the performance of EI in this setting. His work focuses on a link between EI and another algorithm known as the optimal computing budget allocation [3], but his analysis reveals EI allocates a vanishing proportion of samples to suboptimal arms as the total number of samples grows. Any method with this property will be far from optimal in BAI problems [1].

In this paper, we improve the EI algorithm dramatically through a simple modification. The resulting algorithm, which we call *top-two expected improvement* (TTEI), combines the top-two sampling idea of Russo [19] with a careful change to the improvement-measure used by EI. We show that this simple variant of EI achieves strong asymptotic optimality properties in the BAI problem, and benchmark the algorithm in simulation experiments.

Our main theoretical contribution is a complete characterization of the asymptotic proportion of samples TTEI allocates to each arm as a function of the true (unknown) arm means. These particular sampling proportions have been shown to be optimal from several perspectives [4, 12, 9, 19, 8], and this enables us to establish two different optimality results for TTEI. The first concerns the rate at which the algorithm gains confidence about the identity of the optimal arm as the total number of samples collected grows. Next we study the so-called fixed confidence setting, where the algorithm is able to stop at any point and return an estimate of the optimal arm. We show that when applied with the stopping rule of Garivier and Kaufmann [8], TTEI essentially minimizes the expected number of samples required among all rules obeying a constraint on the probability of incorrect selection.

One undesirable feature of our algorithm is its dependence on a tuning parameter. Our theoretical results precisely show the impact of this parameter, and reveal a surprising degree of robustness to its value. It is also easy to design methods that adapt this parameter over time to the optimal value, and we explore one such method in simulation. Still, removing this tuning parameter is an interesting direction for future research.

**Further related literature.** Despite the popularity of EI, its theoretical properties are not well studied. A notable exception is the work of Bull [2], who studies a global optimization problem and provides a convergence rate for EI's expected loss. However, it is assumed that the observations are noiseless. Our work also relates to a large number of recent machine learning papers that try to characterize the sample complexity of the best-arm identification problem [5, 18, 1, 7, 14, 10, 11, 15–17]. Despite substantial progress, matching asymptotic upper and lower bounds remained elusive in this line of work. Building on older work in statistics [4, 12] and simulation optimization [9], recent work of Garivier and Kaufmann [8] and Russo [19] characterized the optimal sampling proportions. Two notions of asymptotic optimality are established: *sample complexity in the fixed confidence setting* and *rate of posterior convergence*. Garivier and Kaufmann [8] developed two sampling rules designed to closely track the asymptotic optimal proportions and showed that, when combined with a stopping rule motivated by Chernoff [4], this sampling rule minimizes the expected number of samples required to guarantee a vanishing threshold on the probability of incorrect selection is satisfied. Russo [19] independently proposed three simple Bayesian algorithms, and proved that each algorithm attains the optimal rate of posterior convergence. TTEI proposed in this paper is conceptually most similar to the top-two value sampling of Russo [19], but it is more computationally efficient.

## 1.1 Main Contributions

As discussed below, our work makes both theoretical and algorithmic contributions.

**Theoretical:** Our main theoretical contribution is Theorem 1, which establishes that TTEI–a simple modification to a popular Bayesian heuristic–converges to the known optimal asymptotic sampling proportions. It is worth emphasizing that, unlike recent results for other top-two sampling algorithms [19], this theorem establishes that the expected time to converge to the optimal proportions is finite, which we need to establish optimality in the fixed confidence setting. Proving this result required substantial technical innovations. Theorems 2 and 3 are additional theoretical contributions. These mirror results in [19] and [8], but we extract minimal conditions on sampling rules that are sufficient to guarantee the two notions of optimality studied in these papers.

**Algorithmic:** On the algorithmic side, we substantially improve a widely used algorithm. TTEI can be easily implemented by modifying existing EI code, but, as shown in our experiments, can offer an order of magnitude improvement. A more subtle point involves the advantages of TTEI over algorithms that are designed to directly target convergence on the asymptotically optimal proportions. In the experiments, we show that TTEI substantially *outperforms an oracle sampling rule* whose sampling proportions directly track the asymptotically optimal proportions. This phenomenon should be explored further in future work, but suggests that

by carefully reasoning about the value of information TTEI accounts for important factors that are washed out in asymptotic analysis. Finally–as discussed in the conclusion–although we focus on uncorrelated priors we believe our method can be easily extended to more complicated problems like that of best-arm identification in linear bandits [22].

## 2 Problem Formulation

Let $A = \{1, \ldots, k\}$ be the set of arms. At each time $n \in \mathbb{N} = \{0, 1, 2, \ldots\}$, an arm $I_n \in A$ is measured, and an independent noisy reward $Y_{n,I_n}$ is observed. The reward $Y_{n,i} \in \mathbb{R}$ of arm $i$ at time $n$ follows a normal distribution $N(\mu_i, \sigma^2)$ with common known variance $\sigma^2$, but unknown mean $\mu_i$. The objective is to allocate measurement effort wisely in order to confidently identify the arm with highest mean using a small number of measurements. We assume that $\mu_1 > \mu_2 > \ldots > \mu_k$. Our analysis takes place in a *frequentist setting*, in which the true means $(\mu_1, \ldots, \mu_k)$ are fixed but unknown. The algorithms we study, however, are Bayesian in the sense that they begin with prior over the arm means and update the belief to form a posterior distribution as evidence is gathered.

**Prior and Posterior Distributions.**   The sampling rules studied in this paper begin with a normally distributed prior over the true mean of each arm $i \in A$ denoted by $N(\mu_{0,i}, \sigma_{0,i}^2)$, and update this to form a posterior distribution as observations are gathered. By conjugacy, the posterior distribution after observing the sequence $(I_0, Y_{0,I_0}, \ldots, I_{n-1}, Y_{n-1,I_{n-1}})$ is also a normal distribution denoted by $N(\mu_{n,i}, \sigma_{n,i}^2)$. The posterior mean and variance can be calculated using the following recursive equations:

$$\mu_{n+1,i} = \begin{cases} (\sigma_{n,i}^{-2}\mu_{n,i} + \sigma^{-2}Y_{n,i})/(\sigma_{n,i}^{-2} + \sigma^{-2}) & \text{if } I_n = i, \\ \mu_{n,i}, & \text{if } I_n \neq i, \end{cases}$$

and

$$\sigma_{n+1,i}^2 = \begin{cases} 1/(\sigma_{n,i}^{-2} + \sigma^{-2}) & \text{if } I_n = i, \\ \sigma_{n,i}^2, & \text{if } I_n \neq i. \end{cases}$$

We denote the posterior distribution over the vector of arm means by

$$\Pi_n = N(\mu_{n,1}, \sigma_{n,1}^2) \otimes N(\mu_{n,2}, \sigma_{n,2}^2) \otimes \cdots \otimes N(\mu_{n,k}, \sigma_{n,k}^2)$$

and let $\theta = (\theta_1, \ldots, \theta_k)$. For example, with this notation

$$\mathbb{E}_{\theta \sim \Pi_n}\left[\sum_{i \in A} \theta_i\right] = \sum_{i \in A} \mu_{n,i}.$$

The posterior probability assigned to the event that arm $i$ is optimal is

$$\alpha_{n,i} \triangleq \mathbb{P}_{\theta \sim \Pi_n}\left(\theta_i > \max_{j \neq i} \theta_j\right). \tag{1}$$

To avoid confusion, we always use $\theta = (\theta_1, \ldots, \theta_k)$ to denote a random vector of arm means drawn from the algorithm's posterior $\Pi_n$, and $\mu = (\mu_1, \ldots, \mu_k)$ to denote the vector of true arm means.

**Two notions of asymptotic optimality.**   Our first notion of optimality relates to the rate of posterior convergence. As the number of observations grows, one hopes that the posterior distribution definitively identifies the true best arm, in the sense that the posterior probability $1 - \alpha_{n,1}$ assigned by the event that a different arm is optimal tends to zero. By sampling the arms intelligently, we hope this probability can be driven to zero as rapidly as possible. Following Russo [19], we aim to maximize the exponent governing the rate of decay,

$$\liminf_{n \to \infty} -\frac{1}{n} \log\left(1 - \alpha_{n,1}\right),$$

among all sampling rules.

The second setting we consider is often called the "fixed confidence" setting. Here, the agent is allowed at any point to stop gathering samples and return an estimate of the identity of the optimal. In addition to a sampling rule, we require a stopping rule that selects a time $\tau$ at which to stop, and

a decision rule that returns an estimate $\hat{I}_\tau$ of the optimal arm based on the first $\tau$ observations. We consider minimizing the average number of observations $\mathbb{E}[\tau_\delta]$ required by an algorithm (that consists of a sampling rule, a stopping rule and a decision rule) guaranteeing a vanishing probability $\delta$ of incorrect identification, i.e., $\mathbb{P}(\hat{I}_{\tau_\delta} \neq 1) \leq \delta$. Following Garivier and Kaufmann [8], the number of samples required scales with $\log(1/\delta)$, and so we aim to minimize

$$\limsup_{\delta \to 0} \frac{\mathbb{E}[\tau_\delta]}{\log(1/\delta)}$$

among all algorithms with probability of error no more than $\delta$. In this setting, we study the performance of sampling rules when combined with the stopping rule studied by Chernoff [4] and Garivier and Kaufmann [8].

## 3   Sampling Rules

In this section, we first introduce the expected improvement algorithm, and point out its weakness. Then a simple variant of the expected improvement algorithm is proposed. Both algorithms make calculations using function $f(x) = x\Phi(x) + \phi(x)$ where $\Phi(\cdot)$ and $\phi(\cdot)$ are the CDF and PDF of the standard normal distribution. One can show that as $x \to \infty$, $\log f(-x) \sim -x^2/2$, and so $f(-x) \approx e^{-x^2/2}$ for very large $x$. One can also show that $f$ is an increasing function.

**Expected Improvement.**   *Expected improvement* [13] is a simple improvement-based sampling rule. The EI algorithm favors the arm that offers the largest amount of improvement upon a target. The EI algorithm measures the arm $I_n = \arg\max_{i \in A} v_{n,i}$ where $v_{n,i}$ is the EI value of arm $i$ at time $n$. Let $I_n^* = \arg\max_{i \in A} \mu_{n,i}$ denote the arm with largest posterior mean at time $n$. The EI value of arm $i$ at time $n$ is defined as

$$v_{n,i} \triangleq \mathbb{E}_{\theta \sim \Pi_n}\left[\left(\theta_i - \mu_{n,I_n^*}\right)^+\right].$$

where $x^+ = \max\{x, 0\}$. The above expectation can be computed analytically as follows,

$$v_{n,i} = \left(\mu_{n,i} - \mu_{n,I_n^*}\right)\Phi\left(\frac{\mu_{n,i} - \mu_{n,I_n^*}}{\sigma_{n,i}}\right) + \sigma_{n,i}\phi\left(\frac{\mu_{n,i} - \mu_{n,I_n^*}}{\sigma_{n,i}}\right) = \sigma_{n,i}f\left(\frac{\mu_{n,i} - \mu_{n,I_n^*}}{\sigma_{n,i}}\right).$$

The EI value $v_{n,i}$ measures the potential of arm $i$ to improve upon the largest posterior mean $\mu_{n,I_n^*}$ at time $n$. Because $f$ is an increasing function, $v_{n,i}$ is increasing in both the posterior mean $\mu_{n,i}$ and posterior standard deviation $\sigma_{n,i}$.

**Top-Two Expected Improvement.**   The EI algorithm can have very poor performance for selecting the best arm. Once the posterior indicates a particular arm is the best with reasonably high probability, EI allocates nearly all future samples to this arm at the expense of measuring other arms. Recently Ryzhov [20] showed that EI only allocates $\mathcal{O}(\log n)$ samples to suboptimal arms asymptotically. This is a severe shortcoming, as it means $n$ must be extremely large before the algorithm has enough samples from suboptimal arms to reach a confident conclusion.

To improve the EI algorithm, we build on the top-two sampling idea in Russo [19]. The idea is to identify in each period the two "most promising" arms based on current observations, and randomize to choose which to sample. A tuning parameter $\beta \in (0, 1)$ controls the probability assigned to the "top" arm. A naive top-two variant of EI would identify the two arms with largest EI value, and flip a $\beta$–weighted coin to decide which to measure. However, one can prove that this algorithm is not optimal for any choice of $\beta$. Instead, what we call the top-two expected improvement algorithm uses a novel modified EI criterion which more carefully accounts for the decision-maker's uncertainty when deciding which arm to sample.

For $i, j \in A$, define $v_{n,i,j} \triangleq \mathbb{E}_{\theta \sim \Pi_n}[(\theta_i - \theta_j)^+]$. This measures the expected magnitude of improvement arm $i$ offers over arm $j$, but unlike the typical EI criterion, this expectation integrates over the uncertain quality of *both arms*. This measure can be computed analytically as

$$v_{n,i,j} = \sqrt{\sigma_{n,i}^2 + \sigma_{n,j}^2}\, f\left(\frac{\mu_{n,i} - \mu_{n,j}}{\sqrt{\sigma_{n,i}^2 + \sigma_{n,j}^2}}\right).$$

TTEI depends on a tuning parameter $\beta > 0$, set to $1/2$ by default. With probability $\beta$, TTEI measures the arm $I_n^{(1)}$ by optimizing the EI criterion, and otherwise it measures an alternative $I_n^{(2)}$ that offers the largest expected improvement on the arm $I_n^{(1)}$. Formally, TTEI measures the arm

$$I_n = \begin{cases} I_n^{(1)} = \arg\max_{i \in A} v_{n,i}, & \text{with probability } \beta, \\ I_n^{(2)} = \arg\max_{i \in A} v_{n,i,I_n^{(1)}}, & \text{with probability } 1 - \beta. \end{cases}$$

Note that $v_{n,i,i} = 0$, which implies $I_n^{(2)} \neq I_n^{(1)}$.

We notice that TTEI with $\beta = 1$ is the standard EI algorithm. Comparing to the EI algorithm, TTEI with $\beta \in (0, 1)$ allocates much more measurement effort to suboptimal arms. We will see that TTEI allocates $\beta$ proportion of samples to the best arm asymptotically, and it uses the remaining $1 - \beta$ fraction of samples for gathering evidence against each suboptimal arm.

## 4    Convergence to Asymptotically Optimal Proportions

For all $i \in A$ and $n \in \mathbb{N}$, we define $T_{n,i} \triangleq \sum_{\ell=0}^{n-1} \mathbf{1}\{I_\ell = i\}$ to be the number of samples of arm $i$ before time $n$. We will show that under TTEI with parameter $\beta$, $\lim_{n\to\infty} T_{n,1}/n = \beta$. That is, the algorithm asymptotically allocates $\beta$ proportion of the samples to true best arm. Dropping for the moment questions regarding the impact of this tuning parameter, let us consider the optimal asymptotic proportion of effort to allocate to each of the $k - 1$ remaining arms. It is known that the optimal proportions are given by the unique vector $(w_2^\beta, \cdots, w_k^\beta)$ satisfying $\sum_{i=2}^{k} w_i^\beta = 1 - \beta$ and

$$\frac{(\mu_1 - \mu_2)^2}{1/\beta + 1/w_2^\beta} = \ldots = \frac{(\mu_1 - \mu_k)^2}{1/\beta + 1/w_k^\beta}. \tag{2}$$

We set $w_1^\beta = \beta$, so $w^\beta = \left(w_1^\beta, \ldots, w_k^\beta\right)$ encodes the sampling proportions of each arm.

To understand the source of equation (2), imagine that over the first $n$ periods each arm $i$ is sampled exactly $w_i^\beta n$ times, and let $\hat{\mu}_{n,i} \sim N\left(\mu_i, \frac{\sigma^2}{w_i^\beta n}\right)$ denote the empirical mean of arm $i$. Then

$$\hat{\mu}_{n,1} - \hat{\mu}_{n,i} \sim N\left(\mu_1 - \mu_i, \tilde{\sigma}_i^2\right) \quad \text{where} \quad \tilde{\sigma}_i^2 = \frac{\sigma^2}{n}\left(\frac{1}{\beta} + \frac{1}{w_i^\beta}\right).$$

The probability $\hat{\mu}_{n,1} - \hat{\mu}_{n,i} \leq 0$–leading to an incorrect estimate of which arm has highest mean–is $\Phi\left((\mu_i - \mu_1)/\tilde{\sigma}_i\right)$ where $\Phi$ is the CDF of the standard normal distribution. Equation (2) is equivalent to requiring $(\mu_1 - \mu_i)/\tilde{\sigma}_i$ is equal for all arms $i$, so the probability of falsely declaring $\mu_i \geq \mu_1$ is equal for all $i \neq 1$. In a sense, these sampling frequencies equalize the evidence against each suboptimal arm. These proportions appeared first in the machine learning literature in [19, 8], but appeared much earlier in the statistics literature in [12], and separately in the simulation optimization literature in [9]. As we will see in the next section, convergence to this allocation is a necessary condition for both notions of optimality considered in this paper.

Our main theoretical contribution is the following theorem, which establishes that under TTEI sampling proportions converge to the proportions $w^\beta$ derived above. Therefore, while the sampling proportion of the optimal arm is controlled by the tuning parameter $\beta$, the remaining $1 - \beta$ fraction of measurement is optimally distributed among the remaining $k - 1$ arms. Such a result was established for other top-two sampling algorithms in [19]. The second notion of optimality requires not just convergence to $w^\beta$ with probability 1, but also a sense in which the expected time until convergence is finite. The following theorem presents such a stronger result for TTEI. To make this precise, we introduce a time after which for each arm, the empirical proportion allocated to it is accurate. Specifically, given $\beta \in (0, 1)$ and $\epsilon > 0$, we define

$$M_\beta^\epsilon \triangleq \inf\left\{N \in \mathbb{N} \,:\, \max_{i \in A} |T_{n,i}/n - w_i^\beta| \leq \epsilon \quad \forall n \geq N\right\}. \tag{3}$$

It is clear that $\mathbb{P}(M_\beta^\epsilon < \infty) = 1$ for all $\epsilon > 0$ if and only if $T_{n,i}/n \to w_i^\beta$ with probability 1 for each arm $i \in A$. To establish optimality in the "fixed confidence setting", we need to prove in addition that $\mathbb{E}[M_\beta^\epsilon] < \infty$ for all $\epsilon > 0$, which requires substantial new technical innovations.

**Theorem 1.** *Under TTEI with parameter $\beta \in (0,1)$, $\mathbb{E}[M_\beta^\epsilon] < \infty$ for any $\epsilon > 0$.*

This result implies that under TTEI, $\mathbb{P}(M_\beta^\epsilon < \infty) = 1$ for all $\epsilon > 0$, or equivalently

$$\lim_{n \to \infty} \frac{T_{n,i}}{n} = w_i^\beta \quad \forall i \in A.$$

### 4.1 Problem Complexity Measure

Given $\beta \in (0,1)$, define the problem complexity measure

$$\Gamma_\beta^* \triangleq \frac{(\mu_1 - \mu_2)^2}{2\sigma^2 \left(1/\beta + 1/w_2^\beta\right)} = \ldots = \frac{(\mu_1 - \mu_k)^2}{2\sigma^2 \left(1/\beta + 1/w_k^\beta\right)},$$

which is a function of the true arm means and variances. This will be the exponent governing the rate of posterior convergence, and also characterizing the average number of samples in the fixed confidence stetting. The optimal exponent comes from maximizing over $\beta$. Let us define $\Gamma^* = \max_{\beta \in (0,1)} \Gamma_\beta^*$ and $\beta^* = \arg\max_{\beta \in (0,1)} \Gamma_\beta^*$ and set

$$w^* = w^{\beta^*} = \left(\beta^*, w_2^{\beta^*}, \ldots, w_k^{\beta^*}\right).$$

Russo [19] has proved that for $\beta \in (0,1)$, $\Gamma_\beta^* \geq \Gamma^* / \max\left\{\frac{\beta^*}{\beta}, \frac{1-\beta^*}{1-\beta}\right\}$, and therefore $\Gamma_{1/2}^* \geq \Gamma^*/2$. This demonstrates a surprising degree of robustness to $\beta$. In particular, $\Gamma_\beta$ is close to $\Gamma^*$ if $\beta$ is adjusted to be close to $\beta^*$, and the choice of $\beta = 1/2$ always yields a 2-approximation to $\Gamma^*$.

## 5  Implied Optimality Results

This section establishes formal optimality guarantees for TTEI. Both results, in fact, hold for any algorithm satisfying the conclusions of Theorem 1, and are therefore of broader interest.

### 5.1  Optimal Rate of Posterior Convergence

We first provide upper and lower bounds on the exponent governing the rate of posterior convergence. The same result has been has been proved in Russo [19] for bounded correlated priors. We use different proof techniques to prove the following result for uncorrelated Gaussian priors.

This theorem shows that no algorithm can attain a rate of posterior convergence faster than $e^{-\Gamma^* n}$ and that this is attained by any algorithm that, like TTEI with optimal tuning parameter $\beta^*$, has asymptotic sampling ratios $(w_1^*, \ldots, w_k^*)$. The second part implies TTEI with parameter $\beta$ attains convergence rate $e^{-n\Gamma_\beta^*}$ and that it is optimal among sampling rules that allocation $\beta$–fraction of samples to the optimal arm. Recall that, without loss of generality, we have assumed arm 1 is the arm with true highest mean $\mu_1 = \max_{i \in A} \mu_i$. We will study the posterior mass $1 - \alpha_{n,1}$ assigned to the event that some other has the highest mean.

**Theorem 2** (Posterior Convergence - Sufficient Condition for Optimality). *The following properties hold with probability 1:*

1. *Under any sampling rule that satisfies $T_{n,i}/n \to w_i^*$ for each $i \in A$,*

$$\lim_{n \to \infty} -\frac{1}{n} \log\left(1 - \alpha_{n,1}\right) = \Gamma^*.$$

   *Under any sampling rule,*

$$\limsup_{n \to \infty} -\frac{1}{n} \log(1 - \alpha_{n,1}) \leq \Gamma^*.$$

2. *Let $\beta \in (0,1)$. Under any sampling rule that satisfies $T_{n,i}/n \to w_i^\beta$ for each $i \in A$,*

$$\lim_{n \to \infty} -\frac{1}{n} \log(1 - \alpha_{n,1}) = \Gamma_\beta^*.$$

*Under any sampling rule that satisfies $T_{n,1}/n \to \beta$,*

$$\limsup_{n\to\infty} -\frac{1}{n}\log(1-\alpha_{n,1}) \le \Gamma_\beta^*.$$

This result reveals that when the tuning parameter $\beta$ is set optimally to $\beta^*$, TTEI attains the optimal rate of posterior convergence. Since $\Gamma_{1/2}^* \ge \Gamma^*/2$, when $\beta$ is set to the default value $1/2$, the exponent governing the convergence rate of TTEI is at least half of the optimal one.

## 5.2 Optimal Average Sample Size

**Chernoff's Stopping Rule.** In the fixed confidence setting, besides an efficient sampling rule, a player also needs to design an intelligent stopping rule. This section introduces a stopping rule proposed by Chernoff [4] and studied recently by Garivier and Kaufmann [8]. This stopping rule makes use of the Generalized Likelihood Ratio statistic, which depends on the current maximum likelihood estimates of all unknown means. For each arm $i \in A$, the maximum likelihood estimate of its unknown mean $\mu_i$ at time $n$ is its empirical mean $\hat{\mu}_{n,i} = T_{n,i}^{-1}\sum_{\ell=0}^{n-1}\mathbf{1}\{I_\ell = i\}Y_{\ell,I_\ell}$ where $T_{n,i} = \sum_{\ell=0}^{n-1}\mathbf{1}\{I_\ell = i\}$. Next we define a weighted average of empirical means of arms $i,j \in A$:

$$\hat{\mu}_{n,i,j} \triangleq \frac{T_{n,i}}{T_{n,i}+T_{n,j}}\hat{\mu}_{n,i} + \frac{T_{n,j}}{T_{n,i}+T_{n,j}}\hat{\mu}_{n,j}.$$

Then if $\hat{\mu}_{n,i} \ge \hat{\mu}_{n,j}$, the Generalized Likelihood Ratio statistic $Z_{n,i,j}$ has the following explicit expression:

$$Z_{n,i,j} \triangleq T_{n,i}d(\hat{\mu}_{n,i},\hat{\mu}_{n,i,j}) + T_{n,j}d(\hat{\mu}_{n,j},\hat{\mu}_{n,i,j})$$

where $d(x,y) = (x-y)^2/(2\sigma^2)$ is the Kullback-Leibler (KL) divergence between Gaussian distributions $N(x,\sigma^2)$ and $N(y,\sigma^2)$. Similarly, if $\hat{\mu}_{n,i} < \hat{\mu}_{n,j}$, $Z_{n,i,j} = -Z_{n,j,i} \le 0$ where $Z_{n,j,i}$ is well defined as above. If either arm has never been sampled before, these quantities are not well defined and we take the convention that $Z_{n,i,j} = Z_{n,j,i} = 0$. Given a target confidence $\delta \in (0,1)$, to ensure that one arm is better than the others with probability at least $1 - \delta$, we use the stopping time

$$\tau_\delta \triangleq \inf\left\{n \in \mathbb{N} : Z_n \triangleq \max_{i\in A}\min_{j\in A\setminus\{i\}} Z_{n,i,j} > \gamma_{n,\delta}\right\}$$

where $\gamma_{n,\delta} > 0$ is an appropriate threshold. By definition, $\min_{j\in A\setminus\{i\}} Z_{n,i,j}$ is nonnegative if and only if $\hat{\mu}_{n,i} \ge \hat{\mu}_{n,j}$ for all $j \in A \setminus \{i\}$. Hence, whenever $\hat{I}_n^* \triangleq \arg\max_{i\in A}\hat{\mu}_{n,i}$ is unique, $Z_n = \min_{j\in A\setminus\{\hat{I}_n^*\}} Z_{n,\hat{I}_n^*,j}$.

Next we introduce the exploration rate for normal bandit models that can ensure to identify the best arm with probability at least $1 - \delta$. We use the following result given in Garivier and Kaufmann [8].

**Proposition 1** (Garivier and Kaufmann [8] Proposition 12). *Let $\delta \in (0,1)$ and $\alpha > 1$. There exists a constant $C = C(\alpha, k)$ such that under any sampling rule, using the Chernoff's stopping rule with the threshold $\gamma_{n,\delta}^\alpha = \log(Cn^\alpha/\delta)$ guarantees*

$$\mathbb{P}\left(\tau_\delta < \infty, \arg\max_{i\in A}\hat{\mu}_{\tau_\delta,i} \ne 1\right) \le \delta.$$

**Sample Complexity.** Garivier and Kaufmann [8] recently provided a general lower bound on the number of samples required in the fixed confidence setting. In particular, they show that for any normal bandit model, under any sampling rule and stopping time $\tau_\delta$ that guarantees a probability of error no more than $\delta$,

$$\liminf_{\delta\to 0} \frac{\mathbb{E}[\tau_\delta]}{\log(1/\delta)} \ge \frac{1}{\Gamma^*}.$$

Recall that $M_\beta^\epsilon$, defined in (3), is the first time after which the empirical proportions are within $\epsilon$ of their asymptotic limits. The next result provides a condition in terms of $M_\beta^\epsilon$ that is sufficient to guarantee optimality in the fixed confidence setting.

**Theorem 3** (Fixed Confidence - Sufficient Condition for Optimality). *Let $\delta, \beta \in (0,1)$ and $\alpha > 1$. Under any sampling rule which, if applied with no stopping rule, satisfies $\mathbb{E}[M_\beta^\epsilon] < \infty$ for all $\epsilon > 0$, using the Chernoff's stopping rule with the threshold $\gamma_{n,\delta}^\alpha = \log(Cn^\alpha/\delta)$ (where $C = C(\alpha, k)$) guarantees*

$$\limsup_{\delta \to 0} \frac{\mathbb{E}[\tau_\delta]}{\log(1/\delta)} \le \frac{1}{\Gamma_\beta^*}.$$

When $\beta = \beta^*$ the general lower bound on sample complexity of $1/\Gamma^*$ is essentially matched. In addition, when $\beta$ is set to the default value $1/2$, the sample complexity of TTEI combined with the Chernoff's stopping rule is at most twice the optimal sample complexity since $1/\Gamma_{1/2}^* \le 2/\Gamma^*$.

## 6 Numerical Experiments

To test the empirical performance of TTEI, we conduct several numerical experiments. The first experiment compares the performance of TTEI with $\beta = 1/2$ and EI. The second experiment compares the performance of different versions of TTEI, top-two Thompson sampling (TTTS) [19], knowledge gradient (KG) [6] and oracle algorithms that know the optimal proportions *a priori*. Each algorithm plays arm $i = 1, \ldots, k$ exactly once at the beginning, and then prescribe a prior $N(Y_{i,i}, \sigma^2)$ for unknown arm-mean $\mu_i$ where $Y_{i,i}$ is the observation from $N(\mu_i, \sigma^2)$. In both experiments, we fix the common known variance $\sigma^2 = 1$ and the number of arms $k = 5$. We consider three instances $[\mu_1, \ldots, \mu_5] = [5, 4, 1, 1, 1], [5, 4, 3, 2, 1]$ and $[2, 0.8, 0.6, 0.4, 0.2]$. The optimal parameter $\beta^*$ equals 0.48, 0.45 and 0.35, respectively.

Recall that $\alpha_{n,i}$, defined in (1), denotes the posterior probability that arm $i$ is optimal. Tables 1 and 2 show the average number of measurements required for the largest posterior probability assigned to some arm being the best to reach a given confidence level $c$, i.e., $\max_i \alpha_{n,i} \ge c$. In a Bayesian setting, the probability of correct selection under this rule is *exactly* $c$. The results in Table 1 are averaged over 100 trials. We see that TTEI with $\beta = 1/2$ outperforms standard EI by an order of magnitude.

Table 1: Average number of measurements required to reach the confidence level $c = 0.95$

|                   | TTEI-1/2 | EI      |
|-------------------|----------|---------|
| $[5, 4, 1, 1, 1]$ | 14.60    | 238.50  |
| $[5, 4, 3, 2, 1]$ | 16.72    | 384.73  |
| $[2, .8, .6, .4, .2]$ | 24.39 | 1525.42 |

The second experiment compares the performance of different versions of TTEI, TTTS, KG, a random sampling oracle (RSO) and a tracking oracle (TO). The random sampling oracle draws a random arm in each round from the distribution $w^*$ encoding the asymptotically optimal proportions. The tracking oracle tracks the optimal proportions at each round. Specifically, the tracking oracle samples the arm with the largest ratio its optimal and empirical proportions. Two tracking algorithms proposed by Garivier and Kaufmann [8] are similar to this tracking oracle. TTEI with adaptive $\beta$ (aTTEI) works as follows: it starts with $\beta = 1/2$ and updates $\beta = \hat{\beta}^*$ every 10 rounds where $\hat{\beta}^*$ is the maximizer of equation (2) based on plug-in estimators for the unknown arm-means. Table 2 shows the average number of measurements required for the largest posterior probability being the best to reach the confidence level $c = 0.9999$. The results in Table 2 are averaged over 200 trials. We see that the performances of TTEI with adaptive $\beta$ and TTEI with $\beta^*$ are better than the performances of all other algorithms. We note that TTEI with adaptive $\beta$ substantially outperforms the tracking oracle.

Table 2: Average number of measurements required to reach the confidence level $c = 0.9999$

|                   | TTEI-1/2 | aTTEI | TTEI-$\beta^*$ | TTTS-$\beta^*$ | RSO    | TO    | KG    |
|-------------------|----------|-------|----------------|----------------|--------|-------|-------|
| $[5, 4, 1, 1, 1]$ | 61.97    | 61.98 | **61.59**      | 62.86          | 97.04  | 77.76 | 75.55 |
| $[5, 4, 3, 2, 1]$ | 66.56    | **65.54** | 65.55      | 66.53          | 103.43 | 88.02 | 81.49 |
| $[2, .8, .6, .4, .2]$ | 76.21 | 72.94 | **71.62**     | 73.02          | 101.97 | 96.90 | 86.98 |

In addition to the Bayesian stopping rule tested above, we have run some experiments with the Chernoff stopping rule discussed in Section 5.2. Asymptotic analysis shows these two rules are

similar when the confidence level $c$ is very high. However, the Chernoff stopping rule appears to be too conservative in practice; it typically yields a probability of correct selection much larger than the specified confidence level $c$ at the expense of using more samples. Since our current focus is on allocation rules, we focus on this Bayesian stopping rule, which appears to offer a more fundamental comparison than one based on ad hoc choice of tuning parameters. Developing improved stopping rules is an important area for future research.

## 7    Conclusion and Extensions to Correlated Arms

We conclude by noting that while this paper thoroughly studies TTEI in the case of uncorrelated priors, we believe the algorithm is also ideally suited to problems with complex correlated priors and large sets of arms. In fact, the modified information measure $v_{n,i,j}$ was designed with an eye toward dealing with correlation in a sophisticated way. In the case of a correlated normal distribution $N(\mu, \Sigma)$, one has

$$v_{n,i,j} = \mathbb{E}_{\theta \sim N(\mu,\Sigma)}[(\theta_i - \theta_j)^+] = \sqrt{\Sigma_{ii} + \Sigma_{jj} - 2\Sigma_{ij}} f\left(\frac{\mu_{n,i} - \mu_{n,j}}{\sqrt{\Sigma_{ii} + \Sigma_{jj} - 2\Sigma_{ij}}}\right).$$

This closed form accommodates efficient computation. Here the term $\Sigma_{i,j}$ accounts for the correlation or similarity between arms $i$ and $j$. Therefore $v_{n,i,I_n^{(1)}}$ is large for arms $i$ that offer large potential improvement over $I_n^{(1)}$, i.e. those that (1) have large posterior mean, (2) have large posterior variance, and (3) are not highly correlated with arm $I_n^{(1)}$. As $I_n^{(1)}$ concentrates near the estimated optimum, we expect the third factor will force the algorithm to experiment in promising regions of the domain that are "far" away from the current-estimated optimum, and are under-explored under standard EI.

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
