[Supplementary Material · TTEI_full_camera_ready.pdf]

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

# A   Notation

We now introduce some important notation that will be used throughout the proof. First, recall we use the notation $\mu_{n,i}$ for the posterior mean of arm $i$, and $\hat{\mu}_{n,i}$ for the empirical mean. The empirical mean is only well defined when $T_{n,i} > 0$, which potentially requires us qualify throughout entire proof that equations are only defined under such circumstances. To avoid this, we take the convention that $\hat{\mu}_{n,i} = 0$ when $T_{n,i} = 0$. The default choice of 0 for the mean is unimportant.

The top two arms identified by TTEI at time $n$ are denote by $I_n^{(1)}$ and $I_n^{(2)}$. We let $I_n^* = \arg\max_{i \in A} \mu_{n,i}$ denote the arm with highest posterior mean and $\hat{I}_n = \arg\max_{i \in A} \hat{\mu}_{n,i}$ denote the arm with highest empirical mean. For much of the proof, $\hat{I}_n^* = I_n^*$, in which case the distinction between these two notations is unimportant.

Next, we define
$$\beta_{\min} \triangleq \min\{\beta, 1 - \beta\} \quad \text{and} \quad \beta_{\max} \triangleq \max\{\beta, 1 - \beta\}$$
and
$$\Delta_{\min} \triangleq \min_{i \neq j} |\mu_i - \mu_j| \quad \text{and} \quad \Delta_{\max} \triangleq \max_{i,j \in A}(\mu_i - \mu_j).$$

Since $\beta \in (0, 1)$, $0 < \beta_{\min} \leq \beta_{\max} < 1$. Because of the assumption that the arm means are unique, we have $\Delta_{\min}, \Delta_{\max} > 0$.

There are two sources of randomness in our problem: the randomness in the observation noise, and the randomness in action selection due to the stochastic nature of the TTEI policy. We introduce two variables $W_1$ and $W_2$, which bound the maximum impact of this randomness. First, define the random variable
$$W_1 \triangleq \max_{n \in \mathbb{N}} \max_{i \in A} \sqrt{\frac{T_{n,i} + 1}{\log(e + T_{n,i})}} \left| \frac{\hat{\mu}_{n,i} - \mu_i}{\sigma} \right|. \tag{4}$$

Ignoring logarithmic factors, for any sample path $\left| \frac{\hat{\mu}_{n,i} - \mu_i}{\sigma} \right|$ is bounded by a term of order $W_1 / \sqrt{T_{n,i}}$. Note that the term $1/\sqrt{T_{n,i}}$ is what would be expected by the central limit theorem.

The second source of randomness is due to the stochastic nature of TTEI. For all $i \in A$ and $n \in \mathbb{N}$, define
$$\psi_{n,i} \triangleq \mathbb{P}(I_n = i | \mathcal{F}_n) \quad \text{and} \quad \Psi_{n,i} \triangleq \sum_{\ell=0}^{n-1} \psi_{\ell,i}.$$
where
$$\mathcal{F}_n = \sigma(I_0, Y_{0,I_0}, \cdots, I_{n-1}, Y_{n-1,I_{n-1}})$$
is the $\sigma$-algebra generated by observations before time $n$. Note that for all $i \in A$, $T_{0,i} = \Psi_{0,i} = 0$. Both $T_{n,i}$ and $\Psi_{n,i}$ measure the effort allocated to arm $i$ before time $n$. Next we introduce the second sample-path dependent variable $W_2$ that measures the difference between two measurements of effort under TTEI. Define
$$W_2 \triangleq \max_{n \in \mathbb{N}} \max_{i \in A} \frac{|T_{n,i} - \Psi_{n,i}|}{\sqrt{(1 + \Psi_{n,i}/\beta_{\min}) \log(e^2 + \Psi_{n,i}/\beta_{\min})}}. \tag{5}$$

Ignoring constants and logarithmic terms, for any sample path $|T_{n,i} - \Psi_{n,i}|$ is bounded by a term of order $W_2/\sqrt{\Psi_{n,i}}$.

Our main results require bounding the expected time for certain events to occur under TTEI. The proof strategy is to bound these times for any sample path in terms of the maximal deviations $W_1$ and $W_2$. We can then appeal to concentration results for $W_1$ and $W_2$ that are established in Section C.

A final piece of important notation pertains to an alternative to the random time $M_\beta^\epsilon$ defined in equation (3) in Section 4. It will often be more convenient to work with the time

$$N_\beta^\epsilon \triangleq \inf\left\{ N \in \mathbb{N} : \max_{i \in A} |\hat{\mu}_{n,i} - \mu_i| \leq \epsilon \text{ and } \max_{i \in A} |T_{n,i}/n - w_i^\beta| \leq \epsilon \quad \forall n \geq N \right\} \tag{6}$$

after which both the sampling proportions and estimated arm means are close to their asymptotic limits.

# B  Proof Outline

The remainder of the appendix is organized as follows.

1. Section C provides some basic tail bounds for Gaussian distributions and the expected improvement measure. The section then shows that all moments of the random variables $W_1$ and $W_2$ are finite. A final part looks at the sufficient condition given in Theorem 3, and shows this condition implies the existence of a time after which the estimated mean of each arm is accurate within any fixed tolerance.

2. Section D provides the proof of Theorem 2, establishing that convergence of sampling proportions to some fixed optimal proportions guarantees the optimal rate of posterior convergence.

3. Section E provides the proof of Theorem 3, a sufficient condition under which optimality in the fixed confidence setting is achieved.

4. Section F proves that TTEI satisfies this sufficient condition for optimality, which immediately establishes Theorem 1.

# C  Concentration and Maximal Inequalities

## C.1  Basic Gaussian Tail Bounds

The following two lemmas are standard results on the tail probabilities of Gaussian distributions. The first tail bound holds for all sub-Gaussian distributions, and is a standard concentration inequality [3].

**Lemma 1.** *Let $x > 0$. If $X \sim N(\mu, \sigma^2)$ then $\mathbb{P}(X \geq \mu + x) \leq e^{-x^2/(2\sigma^2)}$.*

The next lemma shows that the exponent in the upper bound above is sharp. That is $\log \mathbb{P}(X \geq \mu + x) = -\frac{x^2}{2\sigma^2} + o(x)$ as $x \to \infty$. This is perhaps the most basic result in large deviations theory and can be found in textbooks [7] or lecture notes [17]. We often consider limits where the number of measurements tends to zero, and as a result the posterior variance of a Gaussian tends to zero. For this reason, the following alternative writing of this result is more convenient for our purposes.

**Lemma 2.** *Fix constants $x > 0$ and $\mu \in \mathbb{R}$. Let $X_\sigma \sim N(\mu, \sigma^2)$ for all $\sigma > 0$. Then*

$$\lim_{\sigma \to 0} \sigma^2 \log \mathbb{P}(X_\sigma \geq \mu + x) = -\frac{x^2}{2}.$$

## C.2  Properties of the EI Measure

We provide several properties of the function $f(x) = x\Phi(x) + \phi(x)$. The first establishes monotonicity. Then Lemmas 4 and 5 provide upper and lower bounds for $f(\cdot)$ which help us to compare two EI values.

**Lemma 3.** *$f(x)$ is positive and increasing on $\mathbb{R}$.*

*Proof.* This is true since $f'(x) = \Phi(x) \geq 0$ and $\lim_{x \to -\infty} f(x) = 0$. □

**Lemma 4.** *For $x > 0$,*
$$f(-x) < \phi(-x).$$

*Proof.* For $x > 0$, $f(-x) = -x\Phi(-x) + \phi(-x) < \phi(-x)$. □

**Lemma 5.** *For $x \geq 2$,*
$$f(-x) > \frac{1}{x^3}\phi(-x).$$

*Proof.* Let $g(x) = \frac{1}{x}[f(-x) - \frac{1}{x^3}\phi(-x)] = -\Phi(-x) + \frac{1}{x}\phi(-x) - \frac{1}{x^4}\phi(-x)$. We have $g'(x) = (-x^{-2} + x^{-3} + 4x^{-5})\phi(x) = x^{-5}(-x+2)(x^2+x+2)\phi(x)$, which implies that $g(x)$ is decreasing in $[2, \infty)$. We notice that $g(2) > 0$ and $\lim_{x \to \infty} g(x) = 0$, so for $x \geq 2$, $g(x) > 0$. Therefore, for $x \geq 2$, $f(-x) > \frac{1}{x^3}\phi(-x)$. □

## C.3 Maximal Inequalities for the Observation Noise

In our theoretical analysis, we need a bound on the difference between the empirical mean $\hat{\mu}_{n,i}$ and the unknown true mean $\mu_i$ for each arm $i \in A$ at time $n$. Recall the definition

$$W_1 = \max_{n \in \mathbb{N}} \max_{i \in A} \sqrt{\frac{T_{n,i}+1}{\log(e+T_{n,i})}} \left| \frac{\hat{\mu}_{n,i} - \mu_i}{\sigma} \right|.$$

given in equation (4). The next lemma establishes that the moment generating function for $W_1$ always exists, which immediately implies bounds on the tails of its distribution function.

To prove this claim, we will use a standard tool in the analysis of bandit algorithms. Imagine writing code to simulate a sampling rule. One way of writing the code waits to see the action $I_n$ selected by TTEI at time $n$, and then generates the corresponding observation $Y_{n,I_n} \sim N(\mu_{I_n}, \sigma^2)$. A mathematically equivalent way of simulating the system is to generate a collection of latent independent random variables $(X_{n,i})_{n \in \mathbb{N}, i \in A}$ where each $X_{n,i} \sim N(\mu_i, \sigma^2)$. At time $n = 0, 1, \ldots$, the algorithm selects an arm $I_n$, and observes the real valued response $X_{T_{n,I_n}, I_n}$. The proof below works directly with the array of latent variables $(X_{n,i})_{n \in \mathbb{N}, i \in A}$, and then deduces implications for $Y_{n,I_n}$ from this. The notation for $X_{n,i}$ is important for this result, but not used again in subsequent proofs.

**Lemma 6.** *Under any sampling rule, $\mathbb{E}[e^{\lambda W_1}] < \infty$ for all $\lambda > 0$.*

*Proof.* Recall that $\hat{\mu}_{n,i} = T_{n,i}^{-1} \sum_{\ell=0}^{n-1} \mathbf{1}\{I_\ell = i\} Y_{\ell, I_\ell}$ and $T_{n,i} = \sum_{\ell=0}^{n-1} \mathbf{1}\{I_\ell = i\}$. We took the convention that $\hat{\mu}_{n,i} = 0$ when $T_{n,i} = 0$, so equation (4) is always well defined. Let $(X_{n,i})_{n \in \mathbb{N}, i \in A}$ be a collection of latent independent random variables where each $X_{n,i} \sim N(\mu_i, \sigma^2)$. For all $i \in A$, we let $\hat{X}_{0,i} = 0$, and for $n \geq 1$, $\hat{X}_{n,i} = \frac{1}{n} \sum_{\ell=0}^{n-1} X_{\ell,i}$ denote the empirical mean of arm $i$ before time $n$. We will bound

$$\xi_0 \triangleq \max_{n \in \mathbb{N}} \max_{i \in A} \sqrt{\frac{n+1}{\log(e+n)}} \left| \frac{\hat{X}_{n,i} - \mu_i}{\sigma} \right|.$$

When every arm is played infinitely often, $W_1 = \xi_0$. One always has $W_1 \leq \xi_0$, so it is sufficient to bound $\mathbb{E}[e^{\lambda \xi_0}]$. It is also sufficient to bound $\mathbb{E}[e^{\lambda \xi}]$ where

$$\xi \triangleq \max_{n \geq 1} \max_{i \in A} \sqrt{\frac{n+1}{\log(e+n)}} \left| \frac{\hat{X}_{n,i} - \mu_i}{\sigma} \right|.$$

For all $n \geq 1$ and $i \in A$, we define $Z_{n,i} \triangleq \sqrt{n} \left( \frac{\hat{X}_{n,i} - \mu_i}{\sigma} \right)$, and then

$$\xi = \max_{n \geq 1} \max_{i \in A} \sqrt{\frac{n+1}{n \log(e+n)}} |Z_{n,i}|.$$

Each $Z_{n,i} \sim N(0,1)$, and thus by Lemma 1, $Z_{n,i}$ satisfies the tail bound $\mathbb{P}(|Z_{n,i}| \geq z) \leq 2e^{-z^2/2}$ for $z > 0$. Therefore, for all $x \geq 2$

$$\mathbb{P}(\xi \geq 2x) = \mathbb{P}\left( \exists n \geq 1, i \in A : |Z_{n,i}| \geq 2\sqrt{\frac{n \log(e+n)}{n+1}} x \right)$$

$$\leq k \sum_{n=1}^{\infty} \mathbb{P}\left( |Z_{n,i}| \geq 2\sqrt{\frac{n \log(e+n)}{n+1}} x \right)$$

$$\leq 2k \sum_{n=1}^{\infty} \exp\left( -\frac{2n \log(e+n)}{n+1} x^2 \right)$$

$$\overset{(*)}{\leq} 2k \sum_{n=1}^{\infty} \exp\left( -2 \log(e+n) - \frac{n}{n+1} x^2 \right)$$

$$= 2k \sum_{n=1}^{\infty} \left( \frac{1}{e+n} \right)^2 e^{-\frac{n}{n+1} x^2}$$

$$\leq C e^{-x^2/2}$$

where step $(*)$ uses the $ab \geq a + b$ when $a, b \geq 2$ and $C = 2k \sum_{n=1}^{\infty}(e + n)^{-2} < \infty$ is a constant. Then for all $\lambda > 0$,

$$\mathbb{E}\left[e^{\lambda \xi}\right] = \int_{x=1}^{\infty} \mathbb{P}\left(e^{\lambda \xi} \geq x\right) \mathrm{d}x \stackrel{(*)}{=} \int_{u=0}^{\infty} \mathbb{P}\left(e^{\lambda \xi} \geq e^{2\lambda u}\right) 2\lambda e^{2\lambda u} \mathrm{d}u$$

$$= 2\lambda \int_{u=0}^{2} \mathbb{P}\left(\xi \geq 2u\right) e^{2\lambda u} \mathrm{d}u + 2\lambda \int_{u=2}^{\infty} \mathbb{P}\left(\xi \geq 2u\right) e^{2\lambda u} \mathrm{d}u$$

$$\leq (e^{4\lambda} - 1) + 2\lambda C \int_{u=2}^{\infty} e^{-u^2/2} \cdot e^{2\lambda u} \mathrm{d}u \ < \infty$$

where in step $(*)$, we have substituted $x = e^{2\lambda u}$. Hence, for all $\lambda > 0$, $\mathbb{E}\left[e^{\lambda W_1}\right] < \infty$. $\qquad \square$

This result provides a bound for the difference between the empirical mean of an arm and its true unknown mean. Specifically, for all $i \in A$ and $n \in \mathbb{N}$,

$$|\hat{\mu}_{n,i} - \mu_i| \leq \sigma W_1 \sqrt{\frac{\log(e + T_{n,i})}{T_{n,i} + 1}}.$$

### C.4 Maximal Inequalities for the Randomness in the Policy

The previous subsection investigates the accuracy of empirical arm means as a function of the number of times $T_{n,i}$ it has been sampled. However, since TTEI is a randomized policy, it is often more natural to study the total probability $\Psi_{n,i}$ the algorithm assigns to measuring arm $i$ throughout the first $n$ measurements. We now consider the random variable

$$W_2 = \max_{n \in \mathbb{N}} \max_{i \in A} \frac{|T_{n,i} - \Psi_{n,i}|}{\sqrt{(1 + \Psi_{n,i}/\beta_{\min}) \log\left(e^2 + \Psi_{n,i}/\beta_{\min}\right)}}.$$

given in equation (5), which captures the maximum relative deviation between $T_{n,i}$ and $\Psi_{n,i}$. As in Lemma 7, we show the moment generating function of $W_2$ always exists, which immediately implies bounds on the tails of the distribution function of $W_2$.

**Lemma 7.** *Under TTEI with parameter $\beta \in (0, 1)$, $\mathbb{E}[e^{\lambda W_2}] < \infty$ for all $\lambda > 0$.*

*Proof.* Similar to the proof of Lemma 6, it suffices to show $\mathbb{P}(W_2 \geq x) \leq k e^{-x^2/2}$ for all $x \geq 2$. Fix some $i \in A$. Define for each $n \in \mathbb{N}$

$$D_n \triangleq T_{n,i} - \Psi_{n,i} = \sum_{\ell=0}^{n-1} d_\ell$$

where

$$d_n \triangleq \mathbf{1}(I_n = i) - \psi_{n,i} = \mathbf{1}(I_n = i) - \mathbb{P}(I_n = i | \mathcal{F}_n).$$

Then $\mathbb{E}[d_n | \mathcal{F}_n] = 0$ and $D_n$ is a zero mean martingale. Note $\psi_{n,i} \in \{0, \beta, 1 - \beta\}$ almost surely, and set

$$X_n \triangleq \mathbf{1}(\psi_{n,i} > 0)$$

to be the indicator that $i$ is among the top-two at time $n$. We can see that $d_n = X_n d_n$, and so

$$D_n = \sum_{\ell=0}^{n-1} X_\ell d_\ell.$$

Here $\{X_n\}$ is a binary valued process and $d_n$ is a zero-mean process with increments bounded as $|d_n| \leq 1$ almost surely. Each $X_n$ is $\mathcal{F}_n$ measureable while each $d_n$ is $\mathcal{F}_{n+1}$ measurable.

The quadratic variation of $D_n$ is

$$\langle D \rangle_n = \sum_{\ell=0}^{n-1} \mathbb{E}[X_\ell d_\ell^2 | \mathcal{F}_\ell] = \sum_{\ell=0}^{n-1} X_\ell \beta (1 - \beta),$$

so the magnitude of fluctuation of the martingale $D_n$ scales with the number of times that arm $i$ is among the top-two.

There are a number of martingale analogues to the central limit theorem, which suggest that $D_n = O_P\left(\sqrt{\langle D \rangle_n}\right)$. To establish this formally, we apply the theorem of self-normalized martingale processes [23], which bound processes like $D_n / \sqrt{\langle D \rangle_n}$. We will apply a result established in [1]. Because $|d_n| \leq 1$, applying Hoeffding's Lemma implies

$$\mathbb{E}[e^{\lambda d_n} | \mathcal{F}_n] \leq e^{\lambda^2/2}, \qquad \lambda \in \mathbb{R}$$

and so $d_n$ is 1-sub–Gaussian conditioned on $\mathcal{F}_n$. Applying Corollary 8 of [1] implies that for any $\delta > 0$, with probability least $1 - \delta$,

$$|D_n| \leq \sqrt{2\left(1 + \sum_{\ell=0}^{n-1} X_\ell\right) \log\left(\frac{\sqrt{1 + \sum_{\ell=0}^{n-1} X_\ell}}{\delta}\right)}, \qquad \forall n \in \mathbb{N}.$$

Analogously, for any $x \geq 2$ with probability at least $1 - e^{-x^2/2}$,

$$
\begin{aligned}
|D_n| \quad &\leq \quad \sqrt{2\left(1 + \sum_{\ell=0}^{n-1} X_\ell\right) \log\left(\frac{\sqrt{1 + \sum_{\ell=0}^{n-1} X_\ell}}{e^{-x^2/2}}\right)} \\
&= \quad \sqrt{\left(1 + \sum_{\ell=0}^{n-1} X_\ell\right)\left(\log\left(1 + \sum_{\ell=0}^{n-1} X_\ell\right) + x^2\right)} \\
&\leq \quad \sqrt{\left(1 + \sum_{\ell=0}^{n-1} X_\ell\right)\left(\log\left(e^2 + \sum_{\ell=0}^{n-1} X_\ell\right) + x^2\right)} \\
&\leq \quad \sqrt{\left(1 + \sum_{\ell=0}^{n-1} X_\ell\right) \log\left(e^2 + \sum_{\ell=0}^{n-1} X_\ell\right) x^2}
\end{aligned}
$$

for all $n \in \mathbb{N}$, where the last step uses that $ab \geq a + b$ for $a, b \geq 2$. Then for all $x \geq 2$

$$\mathbb{P}\left( \max_{n \in \mathbb{N}} \frac{|D_n|}{\sqrt{\left(1 + \sum_{\ell=0}^{n-1} X_\ell\right) \log\left(e^2 + \sum_{\ell=0}^{n-1} X_\ell\right)}} \geq x \right) \leq e^{-x^2/2}$$

Since $\Psi_{n,i} \geq \beta_{\min} \sum_{\ell=0}^{n-1} X_\ell$, we have shown that for any $i$,

$$\mathbb{P}\left( \max_{n \in \mathbb{N}} \frac{|T_{n,i} - \Psi_{n,i}|}{\sqrt{(1 + \Psi_{n,i}/\beta_{\min}) \log\left(e^2 + \Psi_{n,i}/\beta_{\min}\right)}} \geq x \right) \leq e^{-x^2/2}.$$

Taking a union bound over $i \in A$ implies $\mathbb{P}(W_2 \geq x) \leq k e^{-x^2/2}$ for any $x \geq 2$. $\qquad\square$

This result implies that for all $i \in A$ and $n \in \mathbb{N}$,

$$|T_{n,i} - \Psi_{n,i}| \leq W_2 \sqrt{(1 + \Psi_{n,i}/\beta_{\min}) \log\left(e^2 + \Psi_{n,i}/\beta_{\min}\right)}.$$

## C.5 A Random Time after which All Means and Sampling Proportions are Accurate

As defined in equation (3), $M_\beta^\epsilon$ is is a time after which the proportion of samples $T_{n,i}/n$ of any arm $i$ is within $\epsilon$ of its ideal proportion $w_i^\beta$. We have also defined

$$N_\beta^\epsilon = \inf\left\{N \in \mathbb{N} : \max_{i \in A} |\hat{\mu}_{n,i} - \mu_i| \le \epsilon \text{ and } \max_{i \in A} |T_{n,i}/n - w_i^\beta| \le \epsilon \quad \forall n \ge N\right\},$$

given in equation (6), which is the time after which both the empirical sampling proportion and the empirical mean of each arm is accurate. These two appear to be closely related: for small $\epsilon$ and $n \ge M_\beta^\epsilon$ each arm has been sampled $\Omega(n)$ times, and so we expect the empirical mean is close to the true mean at each arm. Based on this intuition, we obtain the following result.

**Lemma 8.** *Let $\beta \in (0,1)$. Under any sampling rule that satisfies $\mathbb{E}[M_\beta^\epsilon] < \infty$ for any $\epsilon > 0$, $\mathbb{E}[N_\beta^\epsilon] < \infty$ for any $\epsilon > 0$.*

*Proof.* Fix $\beta \in (0,1)$. Let $c_\beta = 0.5 \min_{i \in A} w_i^\beta$. By definition of $M_\beta^{c_\beta}$, for all $n \ge M_\beta^{c_\beta}$ and $i \in A$, we have $T_{n,i}/n \ge w_i^\beta - c_\beta \ge c_\beta$, and thus $T_{n,i} \ge nc_\beta$.

Given $\epsilon > 0$. When

$$\frac{T_{n,i} + 1}{\log(e + T_{n,i})} \ge \left(\frac{\sigma W_1}{\epsilon}\right)^2,$$

we have

$$|\hat{\mu}_{n,i} - \mu_i| \le \sigma W_1 \sqrt{\frac{\log(e + T_{n,i})}{T_{n,i} + 1}} \le \epsilon.$$

For all $n \ge M_\beta^{c_\beta} + \left[4 + (\sigma W_1/\epsilon)^4\right]/c_\beta$ and $i \in A$,

$$T_{n,i} \ge nc_\beta \ge 4 + (\sigma W_1/\epsilon)^4,$$

and thus

$$\frac{T_{n,i} + 1}{\log(e + T_{n,i})} \ge T_{n,i}^{1/2} \ge \left(\frac{\sigma W_1}{\epsilon}\right)^2$$

where the first inequality uses $\log(e + T_{n,i}) \le T_{n,i}^{1/2}$ when $T_{n,i} \ge 4$.

Hence, for $n \ge M_\beta^\epsilon + M_\beta^{c_\beta} + [4 + (\sigma W_1/\epsilon)^4]/c_\beta$,

$$\max_{i \in A} |T_{n,i}/n - w_i^\beta| \le \epsilon \quad \text{and} \quad \max_{i \in A} |\hat{\mu}_{n,i} - \mu_i| < \epsilon,$$

which implies

$$N_\beta^\epsilon \le M_\beta^\epsilon + M_\beta^{c_\beta} + \left[4 + (\sigma W_1/\epsilon)^4\right]/c_\beta.$$

Since $\mathbb{E}[M_\beta^\epsilon]$ for any $\epsilon > 0$ and by Lemma 6, $\mathbb{E}[W_1^4] < \infty$, we have $\mathbb{E}[N_\beta^\epsilon] < \infty$ for any $\epsilon > 0$. $\square$

## D  Proof of Theorem 2

To simplify the presentation, we introduce the following asymptotic notation. We say two real-valued sequences $\{a_n\}$ and $\{b_n\}$ are *logarithmically equivalent* if

$$\lim_{n \to \infty} \frac{1}{n} \log\left(\frac{a_n}{b_n}\right) = 0.$$

We denote this relation by $a_n \doteq b_n$. We can show the following result.

**Lemma 9.**

$$1 - \alpha_{n,1} \doteq \max_{i \ne 1} \mathbb{P}_{\theta \sim \Pi_n} (\theta_i \ge \theta_1).$$

*Proof.* By definition, $\alpha_{n,1} = \mathbb{P}_{\theta \sim \Pi_n} (\theta_1 > \max_{i \neq 1} \theta_i)$, and then

$$1 - \alpha_{n,1} = \mathbb{P}_{\theta \sim \Pi_n} \left( \cup_{i \neq 1} (\theta_i \geq \theta_1) \right).$$

Hence,

$$\max_{i \neq 1} \mathbb{P}_{\theta \sim \Pi_n} (\theta_i \geq \theta_1) \leq 1 - \alpha_{n,1} \leq \sum_{i \neq 1} \mathbb{P}_{\theta \sim \Pi_n} (\theta_i \geq \theta_1) \leq (k-1) \max_{i \neq 1} \mathbb{P}_{\theta \sim \Pi_n} (\theta_i \geq \theta_1)$$

where the second inequality uses the union bound. Using the squeeze theorem, we have

$$1 - \alpha_{n,1} \doteq \max_{i \neq 1} \mathbb{P}_{\theta \sim \Pi_n} (\theta_i \geq \theta_1).$$

$\square$

For each sample path, we can divide the set of all arms $A$ into two subsets:

$$\mathcal{I} \triangleq \{ i \in A : \lim_{n \to \infty} T_{n,i} = \infty \} \quad \text{and} \quad \overline{\mathcal{I}} \triangleq A \setminus \mathcal{I}.$$

$\mathcal{I}$ is the set of arms that receive infinite measurement effort. If $\overline{\mathcal{I}}$ is empty, all arms are sampled infinite times, and we have the following result.

**Lemma 10.** *If $\overline{\mathcal{I}}$ is empty,*

$$\mathbb{P}_{\theta \sim \Pi_n} (\theta_i \geq \theta_1) \doteq \exp \left( \frac{-(\mu_1 - \mu_i)^2}{2\sigma^2 (1/T_{n,1} + 1/T_{n,i})} \right) \quad \forall i \neq 1.$$

*Proof.* Fix $i \neq 1$. For $\theta \sim \Pi_n$, $\theta_i - \theta_1 \sim N \left( -\Delta_n, \tilde{\sigma}_n^2 \right)$ where we define $\Delta_n = \mu_{n,1} - \mu_{n,i}$ and

$$\tilde{\sigma}_n^2 = \sigma_{n,i}^2 + \sigma_{n,1}^2 = \frac{1}{1/\sigma_{0,i}^2 + T_{n,i}/\sigma^2} + \frac{1}{1/\sigma_{0,1}^2 + T_{n,1}/\sigma^2}.$$

By hypothesis, $T_{n,i} \to \infty$ and $T_{n,1} \to \infty$. This implies $\Delta_n \to \mu_1 - \mu_i$ and $\tilde{\sigma}_n \to 0$. Applying Lemma 2 shows

$$\mathbb{P}_{\theta \sim \Pi_n} (\theta_i \geq \theta_1) \doteq \exp \left( -\Delta_n^2 / 2\tilde{\sigma}_n^2 \right).$$

For any two sequences $\{a_n\}$ and $\{b_n\}$, $a_n \doteq b_n$ if $n^{-1} \log(a_n) - n^{-1} \log(b_n) \to 0$. We have

$$\frac{\Delta_n^2}{2\tilde{\sigma}_n^2} = \frac{(\mu_1 - \mu_i)^2}{2\sigma^2 (1/T_{n,1} + 1/T_{n,i})} + o(n)$$

since $T_{n,i} \to \infty$ and $T_{n,1} \to \infty$, and therefore

$$\exp \left( -\Delta_n^2 / 2\tilde{\sigma}_n^2 \right) \doteq \exp \left( \frac{-(\mu_1 - \mu_i)^2}{2\sigma^2 (1/T_{n,1} + 1/T_{n,i})} \right).$$

$\square$

Now we are ready to prove the theorem.

**Theorem 2** (Posterior Convergence - Sufficient Condition for Optimality). *The following properties hold with probability 1:*

1. *Under any sampling rule that satisfies $T_{n,i}/n \to w_i^*$ for each $i \in A$,*

$$\lim_{n \to \infty} -\frac{1}{n} \log \left( 1 - \alpha_{n,1} \right) = \Gamma^*. \tag{7}$$

   *Under any sampling rule,*

$$\limsup_{n \to \infty} -\frac{1}{n} \log(1 - \alpha_{n,1}) \leq \Gamma^*. \tag{8}$$

2. *Let $\beta \in (0,1)$. Under any sampling rule that satisfies $T_{n,i}/n \to w_i^\beta$ for each $i \in A$,*

$$\lim_{n\to\infty} -\frac{1}{n}\log(1-\alpha_{n,1}) = \Gamma_\beta^*.$$

*Under any sampling rule that satisfies $T_{n,1}/n \to \beta$,*

$$\limsup_{n\to\infty} -\frac{1}{n}\log(1-\alpha_{n,1}) \leq \Gamma_\beta^*.$$

*Proof.* We begin by establishing the result in equation (8). Recall $\mathcal{I}$ is the set of arms that are sampled an infinite number of times, and $\overline{\mathcal{I}}$ is its complement. Suppose first that $\overline{\mathcal{I}}$ is nonempty. In this case, we show $\lim_{n\to\infty} \alpha_{n,1} < 1$, which implies (8).

For each $i \in A$, we define

$$\mu_{\infty,i} \triangleq \lim_{n\to\infty} \mu_{n,i} \quad \text{and} \quad \sigma_{\infty,i}^2 \triangleq \lim_{n\to\infty} \sigma_{n,i}^2,$$

so for each $i \in \mathcal{I}$

$$\mu_{\infty,i} = \mu_i \quad \text{and} \quad \sigma_{\infty,i}^2 = 0,$$

while for each $i \in \overline{\mathcal{I}}$, $\sigma_{\infty,i}^2 > 0$.

We let

$$\Pi_\infty \triangleq N(\mu_{\infty,1}, \sigma_{\infty,1}^2) \otimes N(\mu_{\infty,2}, \sigma_{\infty,2}^2) \otimes \cdots \otimes N(\mu_{\infty,k}, \sigma_{\infty,k}^2)$$

denote the limiting posterior distribution, with the understanding that for each $i \in \mathcal{I}$, $N(\mu_{\infty,i}, \sigma_{\infty,i}^2)$ represents a point mass at the true mean $\mu_i$. For each $i \in A$, we define

$$\alpha_{\infty,i} \triangleq \mathbb{P}_{\theta \sim \Pi_\infty}\left(\theta_i > \max_{j\neq i}\theta_j\right).$$

Suppose that $\overline{\mathcal{I}}$ is nonempty. For each $i \in \overline{\mathcal{I}}$, since $\sigma_{\infty,i}^2 > 0$, we have $\alpha_{\infty,i} > 0$, which implies $\alpha_{\infty,1} < 1$. Hence,

$$\lim_{n\to\infty} -\frac{1}{n}\log(1-\alpha_{n,1}) = \lim_{n\to\infty} -\frac{1}{n}\log(1-\alpha_{\infty,1}) = 0.$$

Now suppose that $\overline{\mathcal{I}}$ is empty. By Lemma 10, we have

$$1 - \alpha_{n,1} \doteq \max_{i\neq 1} \mathbb{P}_{\theta\sim\Pi_n}\left(\theta_i \geq \theta_1\right)$$

$$\doteq \max_{i\neq 1}\left\{\exp\left(\frac{-(\mu_1-\mu_i)^2}{2\sigma^2(1/T_{n,1}+1/T_{n,i})}\right)\right\}$$

$$\doteq \exp\left(-n\min_{i\neq 1}\left\{\frac{(\mu_1-\mu_i)^2}{2\sigma^2(n/T_{n,1}+n/T_{n,i})}\right\}\right)$$

where the second equality uses the property that if for each $i = 1,\ldots,m$, $a_{n,i} \doteq b_{n,i}$, then $\max_{i\in\{1,\ldots,m\}} a_{n,i} \doteq \max_{i\in\{1,\ldots,m\}} b_{n,i}$.

Let

$$\Sigma \triangleq \left\{w = (w_1,\ldots,w_k) : \sum_{i=1}^k w_i = 1 \text{ and } w_i \geq 0, \forall i \in A\right\}$$

denote the set of possible proportions allocated to $k$ arms. Russo [24] showed that

$$\Gamma^* = \max_{w\in\Sigma}\min_{i\neq 1}\frac{(\mu_1-\mu_i)^2}{2\sigma^2(1/w_1+1/w_i)},$$

where the maximizer is $w^*$.

Under any sampling rule,

$$1 - \alpha_{n,1} \doteq \exp\left(-n\min_{i\neq 1}\left\{\frac{(\mu_1-\mu_i)^2}{2\sigma^2(n/T_{n,i}+n/T_{n,1})}\right\}\right)$$

$$\geq \exp\left(-n\max_{w\in\Sigma}\min_{i\neq 1}\left\{\frac{(\mu_1-\mu_i)^2}{2\sigma^2(1/w_i+1/w_1)}\right\}\right)$$

$$= \exp\left(-n\Gamma^*\right),$$

which implies

$$\limsup_{n\to\infty} -\frac{1}{n}\log(1-\alpha_{n,1}) \le \Gamma^*.$$

This completes the proof of (8).

These same calculations can be leveraged to establish (7). Under any sampling rule that satisfies $T_{n,i}/n \to w_i^*$ for each $i \in A$,

$$\begin{aligned}
1 - \alpha_{n,1} &\doteq \exp\left(-n\min_{i\neq 1}\left\{\frac{(\mu_1-\mu_i)^2}{2\sigma^2(n/T_{n,1}+n/T_{n,i})}\right\}\right) \\
&\doteq \exp\left(-n\min_{i\neq 1}\left\{\frac{(\mu_1-\mu_i)^2}{2\sigma^2(1/w_1^*+1/w_i^*)}\right\}\right) \\
&= \exp(-n\Gamma^*),
\end{aligned}$$

and thus

$$\limsup_{n\to\infty} -\frac{1}{n}\log(1-\alpha_{n,i}) = \Gamma^*.$$

Given $\beta \in (0,1)$, Russo [24] showed that

$$\Gamma_\beta^* = \max_{w\in\Sigma:w_1=\beta} \min_{i\neq 1} \frac{(\mu_1-\mu_i)^2}{2\sigma^2(1/w_1+1/w_i)},$$

where the maximizer is $w^\beta$. Repeating the same proof given above with minor changes in notation, we can show that under any sampling rule that satisfies $T_{n,i}/n \to w_i^\beta$ for each $i \in A$,

$$\lim_{n\to\infty} -\frac{1}{n}\log(1-\alpha_{n,1}) = \Gamma_\beta^*,$$

and under any sampling rule that satisfies $T_{n,1}/n \to \beta$, we have

$$\limsup_{n\to\infty} -\frac{1}{n}\log(1-\alpha_{n,1}) \le \Gamma_\beta^*.$$

$\square$

# E  Proof of Theorem 3

Intuitively speaking, this theorem works because in the Chernoff's stopping rule, the statistic $Z_n \approx \Gamma_\beta^* n$ as $n$ is large and the threshold $\gamma_{n,\delta}^\alpha = \log(Cn^\alpha/\delta) = \log(1/\delta) + o(n)$, which implies that the stopping time $\tau_\delta$ scales as $\log(1/\delta)/\Gamma_\beta^*$.

We first recall the statement of Theorem 3.

**Theorem 3** (Fixed Confidence - Sufficient Condition for Optimality). *Let $\delta, \beta \in (0,1)$ and $\alpha > 1$. Under any sampling rule which, if applied with no stopping rule, satisfies $\mathbb{E}[M_\beta^\epsilon] < \infty$ for all $\epsilon > 0$, using the Chernoff's stopping rule with the threshold $\gamma_{n,\delta}^\alpha = \log(Cn^\alpha/\delta)$ (where $C = C(\alpha,k)$) guarantees*

$$\limsup_{\delta\to 0} \frac{\mathbb{E}[\tau_\delta]}{\log(1/\delta)} \le \frac{1}{\Gamma_\beta^*}.$$

Our proof appeals to the random time $N_\epsilon^\beta$ defined in equation (6) as

$$N_\epsilon^\beta = \inf\left\{N \in \mathbb{N} : \max_{i\in A}|\hat\mu_{n,i}-\mu_i| \le \epsilon \text{ and } \max_{i\in A}|T_{n,i}/n - w_i^\beta| \le \epsilon \quad \forall n \ge N\right\}.$$

Lemma 8 shows that our hypothesis $\mathbb{E}[M_\beta^\epsilon] < \infty$ for all $\epsilon > 0$ implies $\mathbb{E}[N_\beta^\epsilon] < \infty$ for all $\epsilon > 0$.

The proof relies on a sequence of lemmas. The first shows that when $n$ is large, the statistic $Z_n$ used in the Chernoff's stopping rule is approximately equal to $\Gamma_\beta^* n$.

**Lemma 11.** *Fix any $\zeta > 0$. Under the conditions of Theorem 3, there exists $N$ with $\mathbb{E}[N] < \infty$ such that for all $n \ge N$, $Z_n \ge (\Gamma_\beta^* - \zeta)n$.*

*Proof.* Using the formula for KL divergence $d(x, y) = (x - y)^2/(2\sigma^2)$,

$$Z_n = \min_{i \in A \setminus \{\hat{I}_n^*\}} Z_{n, \hat{I}_n^*, i} = \min_{i \in A \setminus \{\hat{I}_n^*\}} \frac{(\hat{\mu}_{n, \hat{I}_n^*} - \hat{\mu}_{n,i})^2}{2\sigma^2(1/T_{n, \hat{I}_n^*} + 1/T_{n,i})}.$$

By the assumption that the true best arm $1 = \arg \max_{i \in A} \mu_i$ is unique, we can choose sufficiently small $\epsilon > 0$ such that $\hat{I}_n^* = 1$ when $\max_{i \in A} |\hat{\mu}_{n,i} - \mu_i| \leq \epsilon$. Hence, for $n \geq N_\beta^\epsilon$,

$$\frac{Z_n}{n} = \min_{i \in A \setminus \{1\}} \frac{(\hat{\mu}_{n,1} - \hat{\mu}_{n,i})^2}{2\sigma^2(n/T_{n,1} + n/T_{n,i})}. \tag{9}$$

Since $\hat{\mu}_{n,1} - \hat{\mu}_{n,i} \to \mu_1 - \mu_i$ and $n/T_{n,1} + n/T_{n,i} \to 1/\beta + 1/w_i^\beta$ and the right hand side of equation (9) is continuous at the limit

$$\Gamma_\beta^* = \frac{(\mu_1 - \mu_2)^2}{2\sigma^2 \left(1/\beta + 1/w_2^\beta\right)} = \ldots = \frac{(\mu_1 - \mu_k)^2}{2\sigma^2 \left(1/\beta + 1/w_k^\beta\right)},$$

there exists a sufficiently small $\epsilon > 0$ such that when $\max_{i \in A} |\hat{\mu}_{n,i} - \mu_i| \leq \epsilon$ and $\max_{i \in A} |T_{n,i}/n - w_i^\beta| \leq \epsilon$, $Z_n/n \geq \Gamma_\beta^* - \zeta$. We set $N = N_\beta^\epsilon$. Then, $Z_n \geq (\Gamma_\beta^* - \zeta)n$ for all $n \geq N$ and by Lemma 8, $\mathbb{E}[N] = \mathbb{E}[N_\beta^\epsilon] < \infty$. $\qquad\square$

Bounding the first time when $Z_n > \gamma_{n,\delta}^\alpha$ is made more technically challenging by the fact that the threshold used in the Chernoff's stopping rule grows logarithmically with $n$. The next lemma simplifies the proof by lower bounding $Z_n - \gamma_{n,\delta}^\alpha$ by a term without this logarithmic factor.

**Lemma 12.** *Fix any $\zeta > 0$. Under the conditions of Theorem 3 there exists $N$ with $\mathbb{E}[N] < \infty$ such that for all $n \geq N$, $Z_n - \gamma_{n,\delta}^\alpha > (\Gamma_\beta^* - 2\zeta)n - \log(C/\delta)$.*

*Proof.* Fix any $\zeta > 0$. There exists a deterministic time $N_1$ such that

$$\gamma_{n,\delta}^\alpha = \log(C/\delta) + \alpha \log(n) < \log(C/\delta) + \zeta n$$

for all $n \geq N_1$. By Lemma 11, there is a random time $N_2$ with $\mathbb{E}[N_2] < \infty$ such that $n \geq N_2$ implies $Z_n \geq (\Gamma_\beta^* - \zeta)n$. Taking $N = N_1 + N_2$ implies the result. $\qquad\square$

We now complete the proof of Theorem 3.

*Proof of Theorem 3.* By definition, if $Z_n - \gamma_{n,\delta}^\alpha > 0$, then $\tau_\delta \leq n$. Fix any $\zeta \in (0, \Gamma_\beta^*/2)$, by Lemma 12, there exists $N$ with $\mathbb{E}[N] < \infty$ such that

$$Z_n - \gamma_{n,\delta}^\alpha > (\Gamma_\beta^* - 2\zeta)n - \log(C/\delta)$$

for all $n \geq N$. Therefore

$$\tau_\delta \leq \max \left\{ N, \frac{\log(C/\delta)}{(\Gamma_\beta^* - 2\zeta)} \right\} \leq N + \frac{\log(C)}{(\Gamma_\beta^* - 2\zeta)} + \frac{\log(1/\delta)}{(\Gamma_\beta^* - 2\zeta)}.$$

Since $\mathbb{E}[N] < \infty$, we find

$$\limsup_{\delta \to 0} \frac{\mathbb{E}[\tau_\delta]}{\log(1/\delta)} \leq \frac{1}{(\Gamma_\beta^* - 2\zeta)}.$$

Since this inequality holds for arbitrarily small $\zeta > 0$, it implies $\limsup_{\delta \to 0} \frac{\mathbb{E}[\tau_\delta]}{\log(1/\delta)} \leq \frac{1}{\Gamma_\beta^*}$. $\qquad\square$

# F   Results specific to TTEI

In this section, we present theoretical results specific to the proposed TTEI policy. The main challenge is ensuring that $\mathbb{E}[M_\beta^\epsilon]$ is finite where $M_\beta^\epsilon$ is the time after which the empirical proportions are $\epsilon$-accurate. To do this, we present several results for each sample path (up to a set of measure zero), and show that $M_\beta^\epsilon$ depends at most polynomially on $W_1$ and $W_2$. By Lemmas 6 and 7, the expected value of polynomials of $W_1$ and $W_2$ is finite. This ensures that $\mathbb{E}[M_\beta^\epsilon]$ is finite, which immediately establishes that TTEI achieves the sufficient conditions for both notions of optimality.

Consistently throughout the proof we make statements like the following: *For all $\epsilon > 0$, there exists a time $N = \mathsf{poly}(W_1, W_2)$ such that for all $n \geq N$, $\max_{i \in A} |\mu_{n,i} - \mu_i| \leq \epsilon$.* This means that for arbitrarily small $\epsilon$, there is a random time $N$ after which all means are $\epsilon$–accurate. Certainly $N$ may depend on $\epsilon$, as implied by the order in which $\epsilon$ and $N$ are chosen. The expression $N = \mathsf{poly}(W_1, W_2)$ means $N = \mathcal{O}(W_1^{c_1} W_2^{c_2})$ for positive constants $c_1$ and $c_2$ where $(\epsilon, \sigma, k, \mu_1, \ldots, \mu_k, \beta)$ are treated as constants. By Lemmas 6 and 7, this is enough to ensure $\mathbb{E}[N] < \infty$. Throughout the entire proof, the problem parameters $(\sigma, k, \mu_1, \ldots, \mu_k, \beta)$ are fixed. Keeping with the use of $n$ to denote a time period, we typically use $N$ to denote a random time after which an event occurs. We typically use $S$ to denote a random number of samples required, such as $T_{n,i} \geq S \implies |\mu_{n,i} - \mu_i| \leq \epsilon$. We use $s$ as a related dummy variable.

For notational convenience, we assume that TTEI begins with an improper prior such that for each arm $i \in A$, $\sigma_{0,i}^2 = \infty$, and we let each $\mu_{0,i} = 0$. Consequently, if $T_{n,i} = \sum_{\ell=0}^{n-1} \mathbf{1}\{I_\ell = i\} = 0$, $\mu_{n,i} = \mu_{0,i} = 0$ and $\sigma_{n,i} = \sigma_{1,i} = \infty$, and if $T_{n,i} > 0$,

$$\mu_{n,i} = \frac{1}{T_{n,i}} \sum_{\ell=0}^{n-1} \mathbf{1}\{I_\ell = i\} Y_{\ell, I_\ell} \quad \text{and} \quad \sigma_{n,i}^2 = \frac{\sigma^2}{T_{n,i}},$$

so the posterior parameters are identical to the frequentist sample mean and variance under the observations collected so far, and thus the arm $I_n^*$ of highest posterior mean is identical to the arm $\hat{I}_n^*$ of highest empirical mean, i.e. $I_n^* = \hat{I}_n^*$. The difference between these formulas and those under a proper prior wash out rapidly as an arm is sampled, but this choice leads to simpler expression in the proof. In this section, *empirical mean* and *posterior mean* are used interchangeably. Under this improper prior, when $T_{n,i} = 0$, $\sigma_{n,i}^2 = \infty$, and we define $v_{n,i} = \infty$ and $v_{n,i,j} = \infty$ for $j \neq i$. This is the natural definition, owing to the fact that if $X_\sigma \sim N(\mu, \sigma^2)$, $\mathbb{E}[(X_\sigma - x)^+] \to \infty$ as $\sigma \to \infty$. Indeed, in the limit as the prior variance $\sigma_{0,i}^2 \to \infty$, $v_{0,i} \to \infty$ and $v_{0,i,j} \to \infty$ for $j \neq i$.

Finally, rather than use the notation $v_{n,i}$ and $v_{n,i,j}$ introduced in Section 3 for the expected-improvement measures, it is more convenient to work with the notation defined here. Set

$$v_{n,i}^{(1)} \triangleq v_{n,i} \quad \forall i \in A$$

to be the expected improvement measure used in identifying the first arm $I_n^{(1)}$ among the top-two, and

$$v_{n,i}^{(2)} \triangleq v_{n,i,I_n^{(1)}} \quad \forall i \in A$$

to be the second expected improvement measure used in identifying the best alternative $I_n^{(2)}$.

## F.1   A Technical Lemma

The following technical lemma is used in the analysis of TTEI to compare two EI values.

**Lemma 13.** *Fix constants $c_0 > c_1 > 0$ and $a_0, c_2 > 0$. Then for any $a_1, a_2 > 0$, there exists a $X = \mathsf{poly}(a_1, a_2)$ such that for all $x \geq X$,*

$$\exp\left(a_0 x^{c_0} - a_1 x^{c_1}\right) > a_2 x^{c_2}.$$

*Proof.* There exists $X_1 = \mathsf{poly}(a_1)$ such that for all $x \geq X_1$, $a_0 x^{c_0 - c_1} - a_1 > 1$. In addition, there exists $X_2 = \mathsf{poly}(a_2)$ such that for all $x \geq X_2$, $\exp\left(x^{c_1}\right) > a_2 x^{c_2}$. Hence, for all $x \geq X \triangleq \max\{X_1, X_2\}$,

$$\exp\left(a_0 x^{c_0} - a_1 x^{c_1}\right) = \exp\left(x^{c_1}\left(a_0 x^{c_0 - c_1} - a_1\right)\right) \geq \exp\left(x^{c_1}\right) > a_2 x^{c_2}.$$

$\square$

## F.2 Sufficient Exploration of All Arms and Concentration of Empirical Means

In this subsection, we first show that when $n$ is large, every arm is sampled frequently. Then the concentration of each empirical mean can be immediately established by the inequality on the difference between the empirical mean and the unknown true mean in terms of the number of samples.

**Proposition 2.** *There exists $N = \mathsf{poly}(W_1, W_2)$ such that for all $n \geq N$,*

$$T_{n,i} \geq (n/k)^{1/4}, \qquad \forall i \in A.$$

This proposition is established in a sequence of results. The first one shows the relationship between the number of samples allocated to the arm $I_n^*$ with the largest posterior mean and the top arm $I_n^{(1)}$ under TTEI. Notice that these two arms could be identical.

**Lemma 14.** *For arms $I_n^*$ and $I_n^{(1)}$,*

$$T_{n,I_n^*} \geq T_{n,I_n^{(1)}}.$$

*Proof.* We prove this by contradiction. Suppose $T_{n,I_n^*} < T_{n,I_n^{(1)}}$. This implies $I_n^* \neq I_n^{(1)}$ and $\sigma_{n,I_n^*} > \sigma_{n,I_n^{(1)}}$. Since $f(x)$ is positive and increasing, we have

$$v_{n,I_n^*}^{(1)} = \sigma_{n,I_n^*} f(0) > \sigma_{n,I_n^{(1)}} f\left(\frac{\mu_{n,I_n^{(1)}} - \mu_{n,I_n^*}}{\sigma_{n,I_n^{(1)}}}\right) = v_{n,I_n^{(1)}}^{(1)},$$

which contradicts that $I_n^{(1)}$ has the largest EI value. Therefore, $T_{n,I_n^*} \geq T_{n,I_n^{(1)}}$. $\qquad\square$

The next result shows that if arm $I_n^*$ and some other arm $i$ are sampled sufficiently, then there is some gap between their empirical means. This result can be used to bound the EI value of arm $i$ from above.

**Lemma 15.** *There exists $S = \mathsf{poly}(W_1)$ such that for all $s \geq S$, if $T_{n,I_n^*} \geq s$ and $T_{n,i} \geq s$ for some $i \neq I_n^*$,*

$$\mu_{n,I_n^*} - \mu_{n,i} \geq \Delta_{\min}/2.$$

*Proof.* If $T_{n,I_n^*} \geq s$, then by Lemma 6,

$$|\mu_{n,I_n^*} - \mu_{I_n^*}| \leq \sigma W_1 \sqrt{\frac{\log(e + T_{n,I_n^*})}{T_{n,I_n^*} + 1}} \leq \sigma W_1 \sqrt{\frac{\log(e + s)}{s + 1}}$$

where the last inequality is valid because $g(x) = \log(e+x)/(x+1)$ is positive and decreasing on $(0, \infty)$. There exists $S = \mathsf{poly}(W_1)$ such that for all $s \geq S$,

$$\sqrt{\frac{\log(e + s)}{s + 1}} \leq \sqrt{\frac{s^{1/2}}{s + 1}} \leq \frac{\Delta_{\min}}{4\sigma W_1},$$

which leads to

$$|\mu_{n,I_n^*} - \mu_{I_n^*}| \leq \Delta_{\min}/4.$$

Similarly, for all $s \geq S$, if $T_{n,i} \geq s$ for some $i \neq I_n^*$,

$$|\mu_{n,i} - \mu_i| \leq \Delta_{\min}/4.$$

Now we want to show that $\mu_{I_n^*} > \mu_i$. We prove this by contradiction. Suppose $\mu_i \geq \mu_{I_n^*}$. This implies that $\mu_i - \mu_{I_n^*} \geq \Delta_{\min}$ since all arm-means are unique. Then we have

$$\mu_{n,i} - \mu_{n,I_n^*} \geq (\mu_i - \Delta_{\min}/4) - (\mu_{I_n^*} + \Delta_{\min}/4) \geq \Delta_{\min}/2,$$

which contradicts the definition of $I_n^*$. Hence, $\mu_{I_n^*} > \mu_i$, and thus

$$\mu_{n,I_n^*} - \mu_{n,i} \geq (\mu_{I_n^*} - \Delta_{\min}/4) - (\mu_i + \Delta_{\min}/4) \geq \Delta_{\min}/2.$$

$\qquad\square$

The next result shows that under a mild condition, if arm $I_n^{(1)}$ has been sampled sufficiently, then $I_n^{(1)} = I_n^*$, the arm with the highest posterior mean. Note that by Lemma 14, $I_n^*$ has been sampled at least as many times as $I_n^{(1)}$. The mild condition of this proof ensures that the number of samples of $I_n^*$ is not an exponential factor larger than the number of samples of $I_n^{(1)}$, which could lead to a contradiction if $I_n^{(1)} \neq I_n^*$.

**Lemma 16.** *There exists $S = \mathsf{poly}(W_1)$ such that for all $s \geq S$, when $n \leq \exp(\Delta_{\min}^2 s/(4\sigma^2))$, if $T_{n,I_n^{(1)}} \geq s$, then $I_n^{(1)} = I_n^*$.*

*Proof.* $T_{n,I_n^{(1)}} \geq s$ is equivalent to $I_n^{(1)} \in \overline{V_n^s}$. By Lemma 14, $T_{n,I_n^*} \geq T_{n,I_n^{(1)}} \geq s$, which means $I_n^* \in \overline{V_n^s}$. We notice that if $v_{n,I_n^*}^{(1)} > v_{n,i}^{(1)}$ for all $i \in \overline{V_n^s} \setminus \{I_n^*\}$, then $I_n^{(1)} = I_n^*$.

By Lemma 15, there exists $S = \mathsf{poly}(W_1)$ such that for all $s \geq S$, $\mu_{n,I_n^*} - \mu_{n,i} \geq \Delta_{\min}/2$ for each $i \in \overline{V_n^s} \setminus \{I_n^*\}$, and thus

$$v_{n,i}^{(1)} = \sigma_{n,i} f\left(\frac{\mu_{n,i} - \mu_{n,I_n^*}}{\sigma_{n,i}}\right) \leq \frac{\sigma}{s^{1/2}} f\left(\frac{-\Delta_{\min} s^{1/2}}{2\sigma}\right) < \frac{\sigma}{\sqrt{2\pi} s^{1/2}} \exp\left(\frac{-\Delta_{\min}^2 s}{8\sigma^2}\right)$$

where the last inequality uses Lemma 4. When $n \leq \exp(\Delta_{\min}^2 s/(4\sigma^2))$,

$$v_{n,I_n^*}^{(1)} = \sigma_{n,I_n^*} f(0) \geq \frac{\sigma}{\sqrt{2\pi} n^{1/2}} \geq \frac{\sigma}{\sqrt{2\pi} s^{1/2}} \exp\left(\frac{-\Delta_{\min}^2 s}{8\sigma^2}\right) > v_{n,i}^{(1)}.$$

This concludes the proof. $\qquad\square$

Now we need to introduce some further notations. Given $s \geq 0$, we define two sets for all $n \in \mathbb{N}$:

$$U_n^s \triangleq \{i \in A : T_{n,i} < s^{1/2}\}$$

and

$$V_n^s \triangleq \{i \in A : T_{n,i} < s\}.$$

We let $\overline{U_n^s} \triangleq A \setminus U_n^s$ and $\overline{V_n^s} \triangleq A \setminus V_n^s$. It is easy to see the following properties:

1. $U_n^s \subseteq V_n^s$.
2. $U_{n+1}^s \subseteq U_n^s$ and $V_{n+1}^s \subseteq V_n^s$.

With these defined sets, we can divide all arms into three sets: $U_n^s$, $\overline{U_n^s} \setminus \overline{V_n^s}$ and $\overline{V_n^s}$. We can view $U_n^s$ and $\overline{V_n^s}$ as the set of arms that are not well explored and the set of arms that are well explored. Then we will show that under some condition, if there exists some arm that is not well explored, then it cannot happen that TTEI measures two arms $I_n^{(1)}$ and $I_n^{(2)}$ that are both well explored.

**Lemma 17.** *There exists $S = \mathsf{poly}(W_1)$ such that for all $s \geq S$, when $n \leq \exp(\Delta_{\min}^2 s/(4\sigma^2))$, if $U_n^s$ is nonempty, $I_n^{(1)} \in V_n^s$ or $I_n^{(2)} \in V_n^s$.*

*Proof.* We want to argue that if $I_n^{(1)} \notin V_n^s$, there exists some arm $j$ such that $v_{n,j}^{(2)} > v_{n,i}^{(2)}$ for all $i \in \overline{V_n^s}$, which implies $I_n^{(2)} \notin \overline{V_n^s}$, and thus $I_n^{(2)} \in V_n^s$. By Lemmas 16 and 15, there exists $S_1 = \mathsf{poly}(W_1)$ such that for all $s \geq S_1$, when $n \leq \exp(\Delta_{\min}^2 s/(4\sigma^2))$, $I_n^{(1)} = I_n^* \in \overline{V_n^s}$, and for each $i \in \overline{V_n^s} \setminus \{I_n^{(1)}\}$, $\mu_{n,I_n^{(1)}} - \mu_{n,i} \geq \Delta_{\min}/2$.

Notice that by definition, $v_{n,I_n^{(1)}}^{(2)} = 0$. For each $i \in \overline{V_n^s} \setminus \{I_n^{(1)}\}$, we have

$$\sigma_{n,i}^2 + \sigma_{n,I_n^{(1)}}^2 = \frac{\sigma^2}{T_{n,i}} + \frac{\sigma^2}{T_{n,I_n^{(1)}}} \leq \frac{\sigma^2}{s} + \frac{\sigma^2}{s} < \frac{4\sigma^2}{s},$$

which leads to

$$v_{n,i}^{(2)} = \sqrt{\sigma_{n,i}^2 + \sigma_{n,I_n^{(1)}}^2} f\left(\frac{\mu_{n,i} - \mu_{n,I_n^{(1)}}}{\sqrt{\sigma_{n,i}^2 + \sigma_{n,I_n^{(1)}}^2}}\right) < \frac{2\sigma}{s^{1/2}} f\left(\frac{-\Delta_{\min} s^{1/2}}{4\sigma}\right) < \frac{2\sigma}{s^{1/2}} \phi\left(\frac{-\Delta_{\min} s^{1/2}}{4\sigma}\right)$$

$$(10)$$

where the last inequality uses Lemma 4.

For any $j \in U_n^s$ (which is possible since $U_n^s$ is nonempty), we have

$$\mu_{n,j} - \mu_{n,I_n^{(1)}} \geq \mu_j - \sigma W_1 \sqrt{\frac{\log(e + T_{n,j})}{T_{n,j} + 1}} - \mu_{I_n^{(1)}} - \sigma W_1 \sqrt{\frac{\log\left(e + T_{n,I_n^{(1)}}\right)}{T_{n,I_n^{(1)}} + 1}}$$

$$\geq \left(\mu_j - \mu_{I_n^{(1)}}\right) - 2\sigma W_1 \sqrt{\frac{\log(e)}{1}}$$

$$= \left(\mu_j - \mu_{I_n^{(1)}}\right) - 2\sigma W_1$$

$$\geq -\Delta_{\max} - 2\sigma W_1$$

where $\Delta_{\max} = \max_{i,j \in A}(\mu_i - \mu_j)$, and the first inequality uses Lemma 6, and the second inequality is valid because $g(x) = \log(e + x)/(x + 1)$ is positive and decreasing on $(0, \infty)$. In addition,

$$\sigma_{n,j}^2 + \sigma_{n,I_n^{(1)}}^2 \geq \sigma_{n,j}^2 = \frac{\sigma^2}{T_{n,j}} > \frac{\sigma^2}{s^{1/2}}.$$

which leads to

$$v_{n,j}^{(2)} > \frac{\sigma}{s^{1/4}} f\left(\frac{-(\Delta_{\max} + 2\sigma W_1)s^{1/4}}{\sigma}\right).$$

For all $s \geq S_2 \triangleq (2\sigma/\Delta_{\max})^4$, we have

$$\frac{(\Delta_{\max} + 2\sigma W_1)s^{1/4}}{\sigma} > \frac{\Delta_{\max} s^{1/4}}{\sigma} \geq 2,$$

and then by Lemma 5,

$$v_{n,j}^{(2)} > \frac{\sigma}{s^{1/4}} f\left(\frac{-(\Delta_{\max} + 2\sigma W_1)s^{1/4}}{\sigma}\right) > \frac{\sigma^4}{s(\Delta_{\max} + 2\sigma W_1)^3} \phi\left(\frac{-(\Delta_{\max} + 2\sigma W_1)s^{1/4}}{\sigma}\right). \tag{11}$$

By Lemma 13, there exists $S_3 = \mathsf{poly}(W_1)$ such that for all $s \geq S_3$, the right hand side of inequality (11) is larger than the right hand side of inequality (10), which implies $I_n^{(2)} \notin \overline{V_n^s}$, and thus $I_n^{(2)} \in V_n^s$.

Therefore, for $s \geq S \triangleq \max\{S_1, S_2, S_3\}$, when $n \leq \exp(\Delta_{\min}^2 s/(4\sigma^2))$, if $U_n^s$ is nonempty $I_n^{(1)} \in V_n^s$ or $I_n^{(2)} \in V_n^s$. $\qquad\square$

With the above lemma, we are ready to show that when $n$ is large, the set $U_n^s$ of arms that are not well explored is indeed empty.

**Lemma 18.** *There exists $S = \mathsf{poly}(W_1, W_2)$ such that for all $s \geq S$, $U_{\lfloor ks^2 \rfloor}^s$ is empty.*

*Proof.* For notational convenience, we assume $\lfloor ks^2 \rfloor = ks^2$.

When $s$ is large, $ks^2 \leq \exp(\Delta_{\min}^2 s/(4\sigma^2))$. Then Lemma 17 implies that there exists $S_1 = \mathsf{poly}(W_1)$ such that for all $s \geq S_1$, when $n \leq ks^2$, if $U_n^s$ is nonempty, then $I_n^{(1)} \in V_n^s$ or $I_n^{(2)} \in V_n^s$. There exists $S_2 = \mathsf{poly}(W_2)$ such that for all $s \geq S_2$,

$$s^2 \geq ks \quad \text{and} \quad \beta_{\min} s^2 - 2kW_2 \sqrt{(1 + ks^2/\beta_{\min}) \log(e^2 + ks^2/\beta_{\min})} \geq ks$$

where $\beta_{\min} = \min\{\beta, 1 - \beta\} > 0$. The reason to construct the above two inequalities becomes clear in the following analysis.

We consider $s \geq S \triangleq \max\{S_1, S_2\}$, and prove the statement by contradiction. Suppose $U_{ks^2}^s$ is nonempty. Then $U_0^s \supseteq U_1^s \supseteq \ldots \supseteq U_{ks^2-1}^s \supseteq U_{ks^2}^s$ are all nonempty. Since $U_n^s \subseteq V_n^s$ for all $n \in \mathbb{N}$, then we $V_0^s \supseteq V_1^s \supseteq \ldots \supseteq V_{ks^2-1}^s \supseteq V_{ks^2}^s$ are all nonempty. Since $s^2 \geq ks$, at least one arm is measured at least $s$ times before time $s^2$, and thus $\left|V_{s^2}^s\right| \leq k - 1$. Then we want to show that for each $r = 2, 3, \ldots, k$, at least one arm in $V_{(r-1)s^2}^s$ is measured at least $s$ times in periods

$\left[(r-1)s^2, rs^2 - 1\right]$. For each $\ell \in \left[(r-1)s^2, rs^2 - 1\right]$, since $U_\ell^s$ is nonempty, by Lemma 17, we have $I_\ell^{(1)} \in V_\ell^s$ or $I_\ell^{(2)} \in V_\ell^s$, and thus

$$\sum_{i \in V_\ell^s} \psi_{\ell,i} = \sum_{i \in V_\ell^s} \mathbb{P}(I_\ell = i | \mathcal{F}_\ell) \geq \beta_{\min},$$

which leads to

$$\sum_{i \in V_{(r-1)s^2}^s} \psi_{\ell,i} \geq \sum_{i \in V_\ell^s} \psi_{\ell,i} \geq \beta_{\min}$$

where we use $V_{(r-1)s^2}^s \supseteq V_\ell^s$. Hence, we have

$$\sum_{i \in V_{(r-1)s^2}^s} \left(\Psi_{rs^2,i} - \Psi_{(r-1)s^2,i}\right) = \sum_{\ell=(r-1)s^2}^{rs^2-1} \sum_{i \in V_{(r-1)s^2}^s} \psi_{\ell,i} \geq \beta_{\min} s^2.$$

Then by Lemma 7, we have

$$\sum_{i \in V_{(r-1)s^2}^s} \left(T_{rs^2,i} - T_{(r-1)s^2,i}\right)$$

$$\geq \sum_{i \in V_{(r-1)s^2}^s} \left[\Psi_{rs^2,i} - W_2\sqrt{(1 + \Psi_{rs^2,i}/\beta_{\min})\log(e^2 + \Psi_{rs^2,i}/\beta_{\min})}\right]$$

$$- \sum_{i \in V_{(r-1)s^2}^s} \left[\Psi_{(r-1)s^2,i} + W_2\sqrt{(1 + \Psi_{(r-1)s^2,i}/\beta_{\min})\log(e^2 + \Psi_{(r-1)s^2,i}/\beta_{\min})}\right]$$

$$\geq \beta_{\min} s^2 - 2kW_2\sqrt{(1 + ks^2/\beta_{\min})\log(e^2 + ks^2/\beta_{\min})}$$

$$\geq ks$$

where the second inequality uses $\left|V_{(r-1)s^2}^s\right| \leq k$ and $\Psi_{(r-1)s^2,i} \leq \Psi_{rs^2,i} \leq \Psi_{ks^2,i} \leq ks^2$ for any $i \in A$. Hence, for each $r = 2, 3, \ldots, k$, at least one arm in $V_{(r-1)s^2}^s$ is sampled at least $s$ times in periods $\left[(r-1)s^2, rs^2 - 1\right]$, which implies $\left|V_{rs^2}^s\right| \leq \left|V_{(r-1)s^2}^s\right| - 1$. Since $\left|V_{s^2}^s\right| \leq k - 1$, by induction, we have $\left|V_{rs^2}^s\right| \leq k - r$ for each $r = 1, 2, \ldots, k$. Hence, $\left|V_{ks^2}^s\right| = 0$, i.e. $V_{ks^2}^s$ is empty. Since $U_{ks^2}^s \subseteq V_{ks^2}^s$, $U_{ks^2}^s$ is empty, which contradicts the supposition that $U_{ks^2}^s$ is nonempty. Therefore, for all $s \geq S$, $U_{ks^2}^s$ is empty. $\qquad\square$

Now we can complete the proof of Proposition 2.

*Proof of Proposition 2.* By Lemma 18, there exists $S$ such that for all $s \geq S$, $U_{\lfloor ks^2 \rfloor}^s$ is empty. Then for all $n \geq N \triangleq kS^2$, we have $\sqrt{n/k} \geq S$, and thus $U_n^{\sqrt{n/k}}$ is empty, which means

$$T_{n,i} \geq \left(\sqrt{n/k}\right)^{1/2} = (n/k)^{1/4}, \qquad \forall i \in A.$$

$\qquad\square$

We have proved that when $n$ is large, each arm is explored sufficiently often. Then using the bound on the difference between the empirical mean $\mu_{n,i}$ and the unknown true mean $\mu_i$ in terms of $T_{n,i}$, we can formally show the concentration of $\mu_{n,i}$ to $\mu_i$.

**Proposition 3.** *Fix a constant $\epsilon > 0$. There exists $N = \mathsf{poly}(W_1, W_2)$ such that for all $n \geq N$,*

$$|\mu_{n,i} - \mu_i| \leq \epsilon, \qquad \forall i \in A.$$

*Proof.* By Lemma 6, for all $i \in A$ and $n \in \mathbb{N}$,

$$|\mu_{n,i} - \mu_i| \leq \sigma W_1 \sqrt{\frac{\log(e + T_{n,i})}{T_{n,i} + 1}}.$$

By Proposition 2, there exists $N_1 = \text{poly}(W_1, W_2)$ such that for all $n \geq N_1$, for all $i \in A$, $T_{n,i} \geq (n/k)^{1/4}$, and thus

$$|\mu_{n,i} - \mu_i| \leq \sigma W_1 \sqrt{\frac{\log(e + T_{n,i})}{T_{n,i} + 1}} \leq \sigma W_1 \sqrt{\frac{\log(e + (n/k)^{1/4})}{(n/k)^{1/4} + 1}}$$

where the last inequality uses $g(x) = \log(e + x)/(x + 1)$ is positive and decreasing on $(0, \infty)$. There exists $N_2 = \text{poly}(W_1)$ such that for all $n \geq N_2$,

$$\sqrt{\frac{\log(e + (n/k)^{1/2})}{(n/k)^{1/2} + 1}} \leq \sqrt{\frac{2(n/k)^{1/4}}{(n/k)^{1/2} + 1}} \leq \frac{\epsilon}{\sigma W_1}.$$

Then for all $n \geq N \triangleq \max\{N_1, N_2\}$, we have

$$|\mu_{n,i} - \mu_i| \leq \sigma W_1 \frac{\epsilon}{\sigma W_1} = \epsilon, \qquad \forall i \in A.$$

$\square$

Notice that the unknown arm-means are unique, so $\mu_1 > \mu_2 \ldots > \mu_k$. Using Proposition 3 on sufficiently small $\epsilon > 0$, we can show that when $n$ is large, the empirical means are so accurate that $\mu_{n,1} > \mu_{n,2} \ldots > \mu_{n,k}$, which means that the arm with the largest posterior mean $I_n^*$ is arm 1.

### F.3 Tracking the Optimal Asymptotic Proportion allocated to the Best Arm

In this subsection, we show that when the number of arm draws goes large, the empirical proportion allocated to the best arm concentrates to the tuning parameter $\beta$ used in TTEI. To prove this, we will first show that when $n$ is large, the arm with the largest EI value $I_n^{(1)}$ is always arm 1.

**Lemma 19.** *There exists $N = \text{poly}(W_1, W_2)$ such that for all $n \geq N$, $I_n^{(1)} = I_n^* = 1$.*

*Proof.* Using Proposition 3 on $\epsilon = \Delta_{\min}/4$ (where $\Delta_{\min} = \min_{i \neq j} |\mu_i - \mu_j| > 0$), there exists $N_1 = \text{poly}(W_1, W_2)$ such that for all $n \geq N_1$,

$$|\mu_{n,i} - \mu_i| \leq \Delta_{\min}/4, \qquad \forall i \in A,$$

which implies $\mu_{n,1} > \mu_{n,2} > \ldots > \mu_{n,k}$, and thus $I_n^* = 1$. Then for $i \neq I_n^*$, we have

$$\begin{aligned}
\mu_{n,I_n^*} - \mu_{n,i} &= \mu_{n,1} - \mu_{n,i} \\
&\geq \mu_1 - \Delta_{\min}/4 - \mu_i - \Delta_{\min}/4 \\
&= (\mu_1 - \mu_i) - \Delta_{\min}/2 \\
&\geq \Delta_{\min}/2.
\end{aligned}$$

By Proposition 2, there exists $N_2 = \text{poly}(W_1, W_2)$ such that for all $n \geq N_2$, $T_{n,i} \geq (n/k)^{1/4}$ for all $i \in A$. Hence, for all $n \geq \max\{N_1, N_2\}$, for each $i \neq I_n^*$, we have

$$v_{n,i}^{(1)} = \sigma_{n,i} f\left(\frac{\mu_{n,i} - \mu_{n,I_n^*}}{\sigma_{n,i}}\right) \leq \frac{\sigma k^{1/8}}{n^{1/8}} f\left(\frac{-\Delta_{\min} n^{1/8}}{2\sigma k^{1/8}}\right) < \frac{\sigma k^{1/8}}{n^{1/8}} \phi\left(\frac{-\Delta_{\min} n^{1/8}}{2\sigma k^{1/8}}\right) \quad (12)$$

where the last inequalities uses Lemma 4. For arm $I_n^*$, we have

$$v_{n,I_n^*}^{(1)} = \sigma_{n,I_n^*} f(0) = \frac{\sigma}{\sqrt{T_{n,I_n^*}}} \phi(0) \geq \frac{\sigma}{n^{1/2}} \phi(0) \quad (13)$$

There exists a deterministic $N_3$ such that for all $n \geq N_3$, the right hand side of inequality (13) is larger than the right hand side of inequality (12). Therefore, for all $n \geq N \triangleq \max\{N_1, N_2, N_3\}$, $v_{n,I_n^*}^{(1)} > v_{n,i}^{(1)}$ for all $i \neq I_n^*$, which leads to $I_n^{(1)} = I_n^* = 1$. $\square$

This lemma immediately implies that when $n$ is large, $\psi_{n,1} = \mathbb{P}(I_n = 1 | \mathcal{F}_n) = \beta$, and then we can show the concentration of $\Psi_{n,i}/n = \sum_{\ell=0}^{n-1} \psi_{\ell,i}/n$ to $\beta$.

**Lemma 20.** *Fix a constant $\epsilon > 0$. There exists $N = \text{poly}(W_1, W_2)$ such that for all $n \geq N$,*
$$\left| \frac{\Psi_{n,1}}{n} - \beta \right| \leq \epsilon.$$

*Proof.* By Lemma 19, there exists $N_1 = \text{poly}(W_1, W_2)$ such that for all $n \geq N_1$, we have $I_n^{(1)} = 1$, which leads to $\psi_{n,1} = \mathbb{P}(I_n = 1 | \mathcal{F}_n) = \beta$. Then for $n \geq N_1$,

$$
\begin{aligned}
\frac{\Psi_{n,1}}{n} &= \frac{1}{n} \left( \sum_{\ell=0}^{N_1-1} \psi_{\ell,1} + \sum_{\ell=N_1}^{n-1} \psi_{\ell,1} \right) \\
&\leq \frac{1}{n} \left[ \beta_{\max} N_1 + \beta(n - N_1) \right] \\
&= \beta + \frac{(\beta_{\max} - \beta) N_3}{n}
\end{aligned}
$$

where $\beta_{\max} = \max\{\beta, 1 - \beta\}$, and

$$
\begin{aligned}
\frac{\Psi_{n,1}}{n} &= \frac{1}{n} \left( \sum_{\ell=0}^{N_1-1} \psi_{\ell,1} + \sum_{\ell=N_1}^{n-1} \psi_{\ell,1} \right) \\
&\geq \frac{1}{n} \beta(n - N_1) \\
&= \beta - \frac{\beta N_1}{n}.
\end{aligned}
$$

For all $n \geq N_2 \triangleq \beta_{\max} N_1 / \epsilon$, we have $(\beta_{\max} - \beta) N_1 / n < \epsilon$ and $-\beta N_1 / n \geq -\epsilon$. Therefore, for all $n \geq N \triangleq \max\{N_1, N_2\}$, we have $|\Psi_{n,1}/n - \beta| \leq \epsilon$. $\qquad \square$

With this result and Lemma 7 which shows that $T_{n,i}/\Psi_{n,i}$ is close to 1 when $\Psi_{n,i}$ is large, we are ready to show the conconetration of $T_{n,1}/n$ to $\beta$.

**Lemma 21.** *Fix a constant $\epsilon > 0$. There exists $N = \text{poly}(W_1, W_2)$ such that for all $n \geq N$,*
$$\left| \frac{T_{n,1}}{n} - \beta \right| \leq \epsilon.$$

*Proof.* It suffices to prove this only for fixed $\epsilon \in (0, \beta]$ since if $\epsilon > \beta$, we can let $N$ be the satisfactory one associated with the constant $\beta$. By Lemma 20, there exists $N_1 = \text{poly}(W_1, W_2)$ such that for all $n \geq N_1$, $|\Psi_{n,1}/n - \beta| \leq \epsilon/2$, which implies $\Psi_{n,1} \geq (\beta - \epsilon/2)n \geq \beta n/2$. By Lemma 7,

$$|T_{n,1} - \Psi_{n,1}| \leq W_2 \sqrt{(1 + \Psi_{n,i}/\beta_{\min}) \log\left(e^2 + \Psi_{n,1}/\beta_{\min}\right)}$$

If $\Psi_{n,1} \geq \beta_{\min}$, we have $1 + \Psi_{n,i}/\beta_{\min} \leq 3\Psi_{n,i}/\beta_{\min}$ and $\log\left(e^2 + \Psi_{n,i}/\beta_{\min}\right) \leq 3(\Psi_{n,i}/\beta_{\min})^{1/2}$, which leads to

$$|T_{n,i} - \Psi_{n,i}| \leq 3 W_2 (\Psi_{n,i}/\beta_{\min})^{3/4} \leq \frac{3 W_2}{\beta_{\min}} \Psi_{n,i}^{3/4}.$$

Then for all $n \geq N_2 \triangleq \max\{N_1, 2\beta_{\min}/\beta\}$, we have $\Psi_{n,1} \geq \beta n/2 \geq \beta_{\min}$, and thus

$$\left| \frac{T_{n,1}}{\Psi_{n,1}} - 1 \right| \leq \frac{3 W_2/\beta_{\min}}{\Psi_{n,i}^{1/4}} \leq \frac{3 W_2/\beta_{\min}}{(\beta n/2)^{1/4}} \tag{14}$$

where the second inequality uses $\Psi_{n,1} \geq \beta n/2$. There exists $N_3 = \text{poly}(W_2)$ such that for all $n \geq N_3$, the right hand side of inequality (14) is less than $\epsilon/(2\beta + \epsilon)$. Hence, for all $n \geq N \triangleq \max\{N_2, N_3\}$, we have $|\Psi_{n,1}/n - \beta| \leq \epsilon/2$ and $|T_{n,1}/\Psi_{n,1} - 1| < \epsilon/(2\beta + \epsilon)$, and thus

$$\frac{T_{n,1}}{n} < \left(1 + \frac{\epsilon}{2\beta + \epsilon}\right) \frac{\Psi_{n,1}}{n} \leq \frac{2(\beta + \epsilon)}{2\beta + \epsilon} (\beta + \epsilon/2) = \beta + \epsilon$$

and

$$\frac{T_{n,1}}{n} < \left(1 - \frac{\epsilon}{2\beta + \epsilon}\right) \frac{\Psi_{n,1}}{n} \geq \frac{2\beta}{2\beta + \epsilon} (\beta - \epsilon/2) > \beta - \epsilon,$$

which leads to $|T_{n,1}/n - \beta| \leq \epsilon$. $\qquad \square$

### F.4 Tracking the Optimal Asymptotic Proportions Allocated to Each Arm

We can further show that for each arm, the empirical proportion allocated to it concentrates to the optimal asymptotic proportion when the number of arm draws goes large.

**Proposition 4.** *Fix a constant $\epsilon > 0$. There exists $N = \mathsf{poly}(W_1, W_2)$ such that for all $n \geq N$,*

$$\left| \frac{T_{n,i}}{n} - w_i^\beta \right| \leq \epsilon, \qquad \forall i \in A.$$

This proposition is established in a sequence of results. We first introduce some further notations. For all $n \in \mathbb{N}$, we define the under-sampled set:

$$P_n \triangleq \left\{ i \neq 1 \; : \; \frac{T_{n,i}}{n} - w_i^\beta < 0 \right\},$$

where $\left( w_2^\beta, \ldots, w_k^\beta \right)$, defined in equation (2) in Section 4, are the optimal asymptotic proportions of effort allocated to each suboptimal arms. In addition, given $\epsilon > 0$, we define the over-sampled set for all $n \in \mathbb{N}$:

$$O_n^\epsilon \triangleq \left\{ i \neq 1 \; : \; \frac{T_{n,i}}{n} - w_i^\beta > \epsilon \right\}.$$

We have shown that when $n$ is large, the empirical proportion allocated to the best arm is accurate, then intuitively speaking, if there is an suboptimal arm that is over-sampled, there should be an suboptimal arm that is under-sampled. Specifically, we have the following result.

**Lemma 22.** *Fix a constant $\epsilon > 0$. There exists $N = \mathsf{poly}(W_1, W_2)$ such that for all $n \geq N$, if $O_n^\epsilon$ is nonempty, then $P_n$ is nonempty.*

*Proof.* By Lemma 21, there exists $N = \mathsf{poly}(W_1, W_2)$ such that for all $n \geq N$, $|T_{n,1}/n - \beta| \leq \epsilon$, which implies $T_{n,1} \geq \beta - \epsilon$. Then we prove this by contradiction. Suppose $P_n$ is empty. For all $i \neq 1$, we have $T_{n,i}/n \geq w_i^\beta$. Since $O_n^\epsilon$ is nonempty, there exists $j \neq 1$ such that $T_{n,j}/n > w_j^\beta + \epsilon$. Then we have

$$\sum_{i \in A} T_{n,i}/n = T_{n,1}/n + T_{n,j}/n + \sum_{i \neq 1, j} T_{n,i}/n$$
$$> \beta - \epsilon + w_j^\beta + \epsilon + \sum_{i \neq 1, j} w_i^\beta$$
$$= \sum_{i \in A} w_i^\beta = 1,$$

which leads to a contradiction. Hence, $P_n$ is nonempty. $\qquad\square$

Next we will show that when $n$ is large, the best alternative $I_n^{(2)}$ is not from the over-sampled set, or equivalently, for any arm $i$ that is over-sampled, it could not be the best alternative $I_n^{(2)}$.

**Lemma 23.** *Fix a constant $\epsilon > 0$. There exists $N = \mathsf{poly}(W_1, W_2)$ such that for all $n \geq N$, $I_n^{(2)} \notin O_n^\epsilon$.*

*Proof.* We want to argue that when $n$ is sufficiently large, if $O_n^\epsilon$ is nonempty, there exists some arm $j$ such that $v_{n,j}^{(2)} > v_{n,i}^{(2)}$ for all $i \in O_n^\epsilon$, which implies $I_n^{(2)} \notin O_n^\epsilon$. By Lemma 19, there exists $N_1 = \mathsf{poly}(W_1, W_2)$ such that for $n \geq N_2$, $I_n^{(1)} = I_n^* = 1$. Then for $i \neq 1$,

$$v_{n,i}^{(2)} = \sqrt{\sigma_{n,i}^2 + \sigma_{n,1}^2} f\left( \frac{\mu_{n,i} - \mu_{n,1}}{\sqrt{\sigma_{n,i}^2 + \sigma_{n,1}^2}} \right).$$

where $\alpha_{n,i}^2 = \sigma^2/T_{n,i}$ and $\alpha_{n,1}^2 = \sigma^2/T_{n,1}$. By Lemma 22, there exists $N_2 = \mathsf{poly}(W_1, W_2)$ such that for all $n \geq N_1$, if $O_n^\epsilon$ is nonempty, $P_n$ is nonempty. We will show that when $n$ is sufficiently

large, for all $i \in O_n^\epsilon$ and $j \in P_n$, $v_{n,j}^{(2)} > v_{n,i}^{(2)}$. By Proposition 3 and Lemma 21, for any $\zeta > 0$, there exists $N_3 = \text{poly}(W_1, W_2)$ (that depends on $\zeta$) such that for all $n \geq N_3$, $|\mu_{n,i} - \mu_i| < \zeta$ for each $i \in A$ and $|T_{n,1}/n - \beta| \leq \zeta$. We let $\zeta$ be sufficiently small such that $T_{n,1} \geq (\beta - \zeta)n > 0$ and $\mu_{n,i} - \mu_{n,1} \leq \mu_i - \mu_1 + 2\zeta < 0$ for each $i \neq 1$. Then for $i \in O_n^\epsilon$, we have $T_{n,i} > (w_i^\beta + \epsilon)n$, and thus

$$
v_{n,i}^{(2)} < \sigma \left( \frac{1}{w_i^\beta + \epsilon} + \frac{1}{\beta - \zeta} \right)^{1/2} n^{-1/2} \phi \left( \frac{(\mu_i - \mu_1 + 2\zeta)n^{1/2}}{\sigma \left[ 1/(w_i^\beta + \epsilon) + 1/(\beta - \zeta) \right]^{1/2}} \right)
$$

where Lemma 4 is used. For $j \in P_n$, we have $T_{n,j} < w_j^\beta n$, and thus

$$
v_{n,j}^{(2)} > \sigma \left( \frac{1}{w_j^\beta} + \frac{1}{\beta + \zeta} \right)^{1/2} n^{-1/2} f \left( \frac{(\mu_j - \mu_1 - 2\zeta)n^{1/2}}{\sigma \left[ 1/w_j^\beta + 1/(\beta + \zeta) \right]^{1/2}} \right)
$$

$$
> \sigma^4 \left( \frac{1}{w_j^\beta} + \frac{1}{\beta + \zeta} \right)^2 (-\mu_j + \mu_1 + 2\zeta)^{-3} n^{-2} \phi \left( \frac{(\mu_j - \mu_1 - 2\zeta)n^{1/2}}{\sigma \left[ 1/w_j^\beta + 1/(\beta + \zeta) \right]^{1/2}} \right)
$$

where Lemma 5 is used in the last inequality for all $n \geq N_4$ such that the value taken by $f$ is less than $-2$.

Now we need to define some further notations to compare the above two EI values. For any $i, j \neq 1$, as $x \to 0$,

$$
g_{i,j}(x) \triangleq \frac{(\mu_i - \mu_1 + 2x)^2}{1/(w_i^\beta + \epsilon) + 1/(\beta - x)} - \frac{(\mu_j - \mu_1 - 2x)^2}{1/w_j^\beta + 1/(\beta + x)}
$$

is continuous at the limit

$$
C_{i,j} \triangleq \frac{(\mu_i - \mu_1)^2}{1/(w_i^\beta + \epsilon) + 1/\beta} - \frac{(\mu_j - \mu_1)^2}{1/w_j^\beta + 1/\beta},
$$

where $\epsilon > 0$ is fixed, and we let

$$
C_{\min} \triangleq \min_{i,j \neq 1} C_{i,j}.
$$

Since

$$
\frac{(\mu_2 - \mu_1)^2}{1/w_2^\beta + 1/\beta} = \ldots = \frac{(\mu_k - \mu_1)^2}{1/w_k^\beta + 1/\beta},
$$

we have $C_{i,j} > 0$ for any $i, j \neq 1$, and thus $C_{\min} > 0$. Clearly, there exists sufficiently small $\tilde{x} > 0$ such that for any $i, j \neq 1$,

$$
g_{i,j}(\zeta) \geq C_{\min}/2,
$$

and we let $\zeta = \tilde{x}$. Next we define

$$
D_{i,j} \triangleq \frac{\sigma^3 \left[ 1/w_j^\beta + 1/(\beta + \zeta) \right]^2 (-\mu_j + \mu_1 + 2\zeta)^{-3}}{\left[ 1/(w_i^\beta + \epsilon) + 1/(\beta - \zeta) \right]^{1/2}}
$$

for any $i, j \neq 1$, and we let

$$
D_{\min} \triangleq \min_{i,j \neq 1} D_{i,j}.
$$

With these notations, we know that for all $n \geq N_5 \triangleq \max\{N_1, N_2, N_3, N_4\}$, for all $i \in O_n^\epsilon$ and $j \in P_n$,

$$
\frac{v_{n,j}^{(2)}}{v_{n,i}^{(2)}} > D_{i,j} n^{-3/2} \exp \left( \frac{C_{\min} n}{4\sigma^2} \right) \geq D_{\min} n^{-3/2} \exp \left( \frac{C_{\min} n}{4\sigma^2} \right). \tag{15}
$$

There exists $N_6$ such that for all $n \geq N_6$, the right hand side of inequality (15) is greater than 1, which is $v_{n,j}^{(2)} > v_{n,i}^{(2)}$. There, for $n \geq N \triangleq \max\{N_5, N_6\}$, $v_{n,j}^{(2)} > v_{n,i}^{(2)}$ for all $i \in O_n^\epsilon$ and $j \in P_n$, which implies $I_n^{(2)} \notin O_n^\epsilon$. $\qquad\square$

The above result and Lemma 19 together shows that when $n$ is large, for any arm $i \neq 1$ that is over-sampled, $I_n^{(2)} \neq i$ and $I_n^{(1)} = 1 \neq i$, which means that arm $i$ is not sampled at time $n$. Next we are going to show that indeed, there is no over-sampled arm when $n$ is sufficiently large.

**Lemma 24.** *Fix a constant $\epsilon > 0$. There exists $N = \mathsf{poly}(W_1, W_2)$ such that for all $n \geq N$, $O_n^\epsilon$ is empty.*

*Proof.* By Lemma 19, there exists $N_1$ such that for all $n \geq N_1$, $I_n^{(1)} = 1$. By Lemma 23, there exists $N_2 = \mathsf{poly}(W_1, W_2)$ such that for all $n \geq N_2$, if $O_n^{\epsilon/2}$ is nonempty, then $I_n^{(2)} \notin O_n^{\epsilon/2}$. Let $M \triangleq \max\{N_1, N_2, 2/\epsilon\}$. Without loss of generality, we assume $M \in \mathbb{N}$ for notational convenience.

For any $i \notin O_M^{\epsilon/2}$ such that $i \neq 1$, we want to prove by induction that for all $n \geq M$, $T_{n,i}/n - w_i^\beta \leq \epsilon$, i.e. $i \notin O_n^\epsilon$. By definition, $T_{M,i}/M - w_i^\beta \leq \epsilon/2 < \epsilon$. Suppose that for a fixed $n \geq M$, $T_{n,i}/n - w_i^\beta \leq \epsilon$. We want to show that $T_{n+1,i}/(n+1) - w_i^\beta \leq \epsilon$. There are two situations. If $T_{n,i}/n - w_i^\beta \leq \epsilon/2$, then

$$\frac{T_{n+1,i}}{n+1} - w_i^\beta < \frac{1 + T_{n,i}}{n} - w_i^\beta \leq 1/n + \epsilon/2 \leq \epsilon$$

where the last inequality uses $n \geq M \geq 2/\epsilon$. On the other hand, if $\epsilon/2 < T_{n,i}/n - w_i^\beta \leq \epsilon$, then by Lemma 23, $I_n^{(2)} \neq i$, and we also have $I_n^{(1)} = 1 \neq i$. Hence,

$$\frac{T_{n+1,i}}{n+1} - w_i^\beta = \frac{T_{n,i}}{n+1} - w_i^\beta \leq \frac{T_{n,i}}{n} - w_i^\beta \leq \epsilon.$$

Combining these two situation, we have $T_{n+1,i}/(n+1) - w_i^\beta \leq \epsilon$. Therefore, by induction, for all $n \geq M$, $T_{n,i}/n - w_i^\beta \leq \epsilon$, i.e. $i \notin O_n^\epsilon$.

For any $i \in O_M^{\epsilon/2}$, if we can find $M_i \geq M$ such that $i \notin O_{M_i}^{\epsilon/2}$ is empty, then same as the proof above, we can show that for all $n \geq M_i$, $T_{n,i}/n - w_i^\beta \leq \epsilon$, i.e. $i \notin O_n^\epsilon$. By definition, $T_{M,i}/M - w_i^\beta > \epsilon/2$. Notice that for all $n \geq M$, $I_n^{(1)} = 1 \neq i$ and by Lemma 23, if $T_{n,i}/n - w_i^\beta > \epsilon/2$, $I_n^{(2)} \neq i$. Hence, arm $i$ is not sampled until the empirical proportion allocated to it is less than or equal to $w_i^\beta + \epsilon/2$. There exists $M_i$ such that for all $n \in [M, M_i - 1]$,

$$\frac{T_{n,i}}{n} - w_i^\beta = \frac{T_{M,i}}{n} - w_i^\beta > \epsilon/2$$

and

$$\frac{T_{M_i,i}}{M_i} - w_i^\beta \leq \epsilon/2,$$

which is $i \notin O_{M_i}^{\epsilon/2}$. Hence, for all $n \geq M_i$, $i \notin O_n^\epsilon$.

Let $N = M/\min_{j \in A} w_j^\beta$. Without loss of generality, we assume $N \in \mathbb{N}$ for notational convenience. Now we want to prove that each $M_i \leq N$. Suppose $M_i \geq N + 1$. Then $N \in [M, M_i - 1]$, so

$$\frac{T_{N,i}}{N} - w_i^\beta = \frac{T_{M,i}}{N} - w_i^\beta > \epsilon/2$$

However, we also have

$$\frac{T_{N,i}}{N} - w_i^\beta = \frac{T_{M,i}}{N} - w_i^\beta \leq \frac{M}{N} - w_i^\beta = \min_{j \in A} w_j^\beta - w_i \leq 0,$$

which leads to a contradiction. Hence, $M_i \leq N$.

Since $M = \mathsf{poly}(W_1, W_2)$, $N = M/\min_{j \in A} w_j^\beta = \mathsf{poly}(W_1, W_2)$, and for all $n \geq N$, $i \notin O_n^\epsilon$ for all $i \neq 1$, which means $O_n^\epsilon$ is empty. $\square$

We can further show that when $n$ is large, the under-sampled set is also empty, which immediately establishes Proposition 4.

*Proof of Proposition 4.* By Lemmas 21 and 24, there exists $N = \mathsf{poly}(W_1, W_2)$ such that for all $n \geq N$,

$$|T_{n,1}/n - w_1^\beta| \leq \epsilon/k \quad \text{and} \quad T_{n,i}/n - w_i^\beta \leq \epsilon/k, \quad \forall i \neq 1$$

where $w_1^\beta = \beta$. Suppose there exists $j \in A$ such that $T_{n,j}/n - w_j^\beta < -\epsilon$. Then

$$\begin{aligned}
\sum_{i \in A} T_{n,i}/n &= T_{n,j}/n + \sum_{i \neq j} T_{n,i}/n \\
&< w_j^\beta - \epsilon + \sum_{i \neq i'} (w_i^\beta + \epsilon/k) \\
&= \sum_{i \in A} w_i^\beta - \epsilon/k = 1 - \epsilon/k,
\end{aligned}$$

which contradicts $\sum_{i \in A} T_{n,i}/n = 1$. Hence, for all $n \geq N$, we also have

$$T_{n,i}/n - w_i^\beta \geq -\epsilon, \qquad \forall i \in A.$$

This concludes the proof. $\qquad\square$

### F.5 Proof of Theorem 1

Based on Proposition 4, Theorem 1 can be immediately established.

**Theorem 1.** *Under TTEI with parameter $\beta \in (0,1)$, $\mathbb{E}[M_\beta^\epsilon] < \infty$ for any $\epsilon > 0$.*

*Proof.* For any $\epsilon > 0$,

$$M_\beta^\epsilon = \inf \left\{ N \in \mathbb{N} : \max_{i \in A} |T_{n,i}/n - w_i^\beta| \leq \epsilon \quad \forall n \geq N \right\}.$$

By Proposition 4, there exists $N = \mathsf{poly}(W_1, W_2)$ such that for all $n \geq N$, $\max_{i \in A} |T_{n,i}/n - w_i^\beta| \leq \epsilon$, which implies $M_\beta^\epsilon \leq N$. By Lemmas 6 and 7, we have $\mathbb{E}[e^{\lambda W_1}] < \infty$ and $\mathbb{E}[e^{\lambda W_2}] < \infty$ for all $\lambda > 0$, which implies $\mathbb{E}[\mathsf{poly}(W_1, W_2)] < \infty$. Therefore, $\mathbb{E}[M_\beta^\epsilon] \leq \mathbb{E}[N] < \infty$. $\qquad\square$