[Reviews · NeurIPS 2017]

Reviewer 1



In this paper the authors propose an optimal algorithm for the best arm identification problem, where the rewards of arms are drawn from normal distributions with unknown means and with a common known variance. The algorithm called Top-Two Expected Improvement (TTEI) is inspired by Expected Improvement algorithm, by the recent work of Russo [19] on top two-sampling and those of Garivier and Kauffman [8]on Chernoff’s stopping rule. The Expected Improvement criterion is modified in order to take into account the expected improvement of an arm over another one. Then the top two sampling idea is used to choose two arms: the first is chosen using the Expected Improvement criterion, and the second one is chosen using the modified criterion with respect to the first chosen arm. A coin is tossed to choose the first arm with a probability \beta, and the second with a probability 1-\beta. \beta is the parameter of TTEI algorithm. This smart algorithm can be used for the two settings of best arm identification problem: fixed confidence or fixed budget. In the case of fixed confidence the Chernoff’s stopping rule is used. The analysis establishes formal guarantees of optimality for algorithms such as TTEI, which tend to sample the arms with respect to the optimal proportions. Another strong point, the analysis reveals that even if \beta is far from its optimal value (in worst case \beta=1/2 and \beta^*=1 or 0), the sample complexity bound is at most twice the optimal one. The weak point of TTEI algorithm is that it is specialized on one distribution of the exponential family (the normal distribution), while the works of [19,8] handle all exponential distributions with optimal algorithms. The experiments done on normal random variables show a small improvement in comparison to TTTS [19] and a more substantial improvement in comparison to TO [8]. The paper will be stronger, if the analysis of TTEI provided evidences (or clues) on the fact that TTEI performs better on Normal distribution than TO [8] as expected. Also, it would be interesting to test and compare TTEI with state-of-the-art on other distributions of the exponential family. Overall, it is a very good paper. I vote for acceptance.

Reviewer 2



The authors propose a new best arm identification algorithm called TTEI that is an extension of "expected improvement" (EI), a known heuristic from the Bayesian optimization community. The motivation for the extension is to match the asymptotic optimalities (rate of posterior convergence & expected stopping time), which are important properties of an algorithm. The authors analyze the asymptotic property of TTEI, show its (near)-optimality, and present its superior empirical performance in synthetic datasets. Overall, well-written paper on an interesting problem. I highly rate the novelty of the algorithm. On the other hand, the asymptotic guarantee often does not capture the reality well in my opinion (described later, though I think this isn't a reason for rejection). I think the experiment part is the weakest in that they do not describe how and why their algorithm work better. Also, I would like to see how much TTEI behaves differently from TTTS since they share many asymptotic properties. This was not captured in the experiments. Question on the rate of posterior convergence: The authors claim at line 97 that the analysis takes place in a frequentist setting. That is true, but isn't the analysis on posterior convergence essentially on the posterior probability "computed by the algorithm"? Sure, we can talk about a quantity that is computed by the algorithm, but if one considers a frequentist setting then what is the meaning of Theorem 2 and how it is solving the problem in the frequentist setting? That is, shouldn't one also talk about what it means to have (1-\alpha_{n,1}) approach to 0 in the frequentist setting? One weakness would be that the guarantee is limited to Gaussian (unlike other papers like the lil' ucb paper that applies to sub-Gaussian RVs), so it does not apply to bernoulli case that is very popular in practice. I wonder if one can relax the random variables to be sub-Gaussian whose scale parameter is bounded below a known constant. What is the authors' opinion on this? I wonder if there are critical steps in the proof that works for Guassian but not for generic sub-Gaussian. Any moderate-confidence regime results? Recent analysis on the top arm identification (such as Chen and Li On the Optimal Sample Complexity for Best Arm Identification, 2016) tells us that there are some important terms that does not depend on \log(1/\dt), so there might be some practically relevant terms the asymptotic analysis is missing. I appreciate the asymptotic optimality, but what happens in practice is that no one wants to use \delta that is too small and in most cases \log(1/\delta) can be treated as constants in the set of problems we care about. I also wonder whether this fact can affect the practical performance. Still, I like to emphasize that I don't think the lack of non-asymptotic analysis should be the reason for rejection in any ways as such analysis is often hard and I believe people should go beyond that to explore new algorithms. Could you provide a more detailed comparison between Russo'16? Are there any difference in the asymptotic optimality? Any different sampling behavior that one can observe in practice? Below L194, why do we need |mu_n,i - \mu_i| \le \epsilon there? If so, what is the key mathematical reason for that? (or, is it for convenience of mathematical derivation?) As far as intuition goes, it seems like if N is large and the sampling proportion is controlled, then one should be able to control |mu_n,i - \mu_i|. Let me know what the authors think. In experiments, why do we use "max_i \alpha_{n,i} \ge c" as the stopping criterion? I thought the good stopping criterion is the one from Garivier & Kaufmann. How is this stopping rule used for the experiments justified in the frequentist setting? Also, I think the authors must have compared the actual probability of misidentification, especially as a graph in P(misid.) vs the # of samples, which measures 'anytime' performance that I think is practically relevant. Could the authors comment on this? Also, I was expecting the reason why we see RSO works worse than TTEI. L274: f => of L219: allocation => allocate L260: guarantees -> guarantee L294: what is the definition of the bold-faced numbers in Table 2? Does that coloring involves some statistical significancy test?

Reviewer 3



This paper studies the problem of best arm identification for stochastic bandits with a finite number of Gaussian arms. The asymptotic as well as fixed-confidence settings are considered. The author(s) introduce(s) a refined (randomised) version (TTEI) of the Bayesian "Expected Improvement" algorithm (EI) and show that, in contrast to the original algorithm, it is optimal in the sense that all arms have optimal asymptotic sampling ratios (asymptotic setting) and essentially optimal sampling complexity (fixed-confidence setting). Moreover, the paper includes some experimental results comparing the empirical performance of TTEI with that of other algorithms suited for the problem and mentions possible future directions for investigating the problem. I consider this to be an outstanding paper. It is truly a pleasure to read since the author(s) manage(s) to offer a well-organised, lucid and honest account although the topic is fairly complicated. This applies in particular to the long and difficult proofs. I would like to make a few minor remarks that might be worth considering, but are not at all meant as criticism: - The assumption that all arms have common known variance seems to be very optimistic. I would appreciate a comment on how this assumption can be relaxed or when it is realistic. - Since $\beta$ is introduced as a tuning parameter of TTEI, it is not immediately clear why this parameter appears in the general optimal sampling proportions, see e.g. the first paragraph on page 4. Similarly, I feel it is slightly misleading that the proof of Theorem 3 begins with a remark on TTEI although the statement refers to arbritrary algorithms.